# IMPROVING ONLINE REINFORCEMENT LEARNING VIA BEHAVIOR PRIOR DISTILLATION

## ABSTRACT

Existing behavior prior reinforcement learning (BPRL) algorithms predominantly rely on offline pre-training, where a behavior cloning model is learned from offline datasets, and policy priors are used to guide the online fine-tuning of the agent. However, the limited quality of offline datasets often hinders the ability to provide high-value policies that can effectively guide policy updates. The absence of expert trajectories significantly impairs online policy learning, leading to low sample efficiency and suboptimal performance. To address these challenges, we depart from conventional behavior prior approaches and propose a Bidirectional Behavior Prior Distillation (B2PD) algorithm. B2PD leverages action-value priors to guide a conditional variational autoencoder (CVAE) in generating a high-value behavior support set. The resulting expert behavior priors are further distilled into the agent, effectively reducing inefficient exploration and enabling stable policy optimization, while establishing a bidirectional knowledge flow mechanism. Experimental results across both state- and pixel-based environments demonstrate that B2PD significantly improves both sample efficiency and overall performance.

## 1 INTRODUCTION

Online reinforcement learning (RL) is notoriously sample inefficient, particularly compared to more offline paradigms, such as imitation learning (Agrawal et al., 2025; Jia et al., 2025). One potential explanation lies in the reliance on the Bellman equation in most online RL algorithms, which often leads to an ineffective exploration problem (Quang & Lauw, 2024b) in Q-learning due to epistemic uncertainty (Van der Vaart et al., 2025). To tackle the challenge of ineffective exploration in online learning and unstable policy improvement, behavior prior reinforcement learning (BPRL) (Singh et al., 2020; Pertsch et al., 2021; Tirumala et al., 2022) has emerged as a prominent research direction. As a generalization of standard online RL, BPRL first pretrains a behavior cloning model on offline expert datasets (Guo et al., 2024), which serves as a teacher network for policy distillation to guide policy updates, thereby facilitating more efficient policy improvement (Daoudi et al., 2024).

In existing BPRL algorithms, offline pretraining plays a crucial role in accelerating policy convergence. Specifically, it utilizes a behavior cloning model to provide policy priors, which are then distilled into the agent to improve sample efficiency (Ball et al., 2023; Wagenmaker & Pacchiano, 2023) and expedite policy convergence. Although offline pretraining has achieved considerable success within the framework of BPRL, it remains subject to a critical limitation: The generated actions are constrained to samples drawn from static offline datasets, thereby restricting policy quality to that of the pre-collected trajectories (Hao et al., 2023; Xudong et al., 2024b; Chemingui et al., 2025). Although some researchers have employed advantage-weighted learning to mitigate the impact of low-quality samples (Chen et al., 2022; Qing et al., 2024), these approaches still fundamentally rely on the fixed support of offline data.

Moving beyond pre-collected data, online distillation (Li & Jin, 2022; Michel et al., 2023) methods dynamically provide policy priors by maintaining a policy support set, without requiring pre-collected offline data. However, the limited diversity of the prior over the policy support set restricts the performance of the policy in turn. This limitation significantly hinders the broader applicability of such methods in RL. The suboptimality of the policy priors introduces two core challenges: First, it hinders the generation of high-value action priors, which could otherwise facilitate stable policy updates for the agent. Second, it impairs the ability of the policy to provide informative gradi-

ents for the agent, thereby weakening its capacity to correct suboptimal gradient directions induced by Q-guidance. To address these challenges, we propose an online policy distillation method that harnesses advances in generative modeling to actively generate policy priors. While prior works have explored using VAEs (Kingma & Welling, 2013) to reconstruct offline data for rapid policy transfer (Rana et al., 2023; Wilcoxson et al., 2024), leveraging generative models to enhance policy updates in online RL remains underexplored, to the best of our knowledge.

In this work, we seek to answer a fundamental question: Can existing knowledge distillation methods be directly applied to online RL training to incorporate policy priors, without relying on offline pretraining or explicit historical trajectory constraints (Goyal et al., 2022; Ran et al., 2023; Li et al., 2023; Lyu et al., 2025), thereby enabling stable policy improvement?

To explore this, we proactively generate behavioral policy priors and introduce a loss function guided by action values to train a behavior cloning model that improves policy quality. Building on this, we distill expert-level behavior priors into the reinforcement learning agent to enable more efficient policy improvement, as illustrated in Figure 1. Empirically, B2PD delivers substantial gains on standard online RL benchmarks, highlighting its practical effectiveness. Moreover, B2PD is broadly compatible with existing off-policy and online RL algorithms and can be integrated with methods such as SAC (Haarnoja et al., 2018) and TD3 (Fujimoto et al., 2018) with only

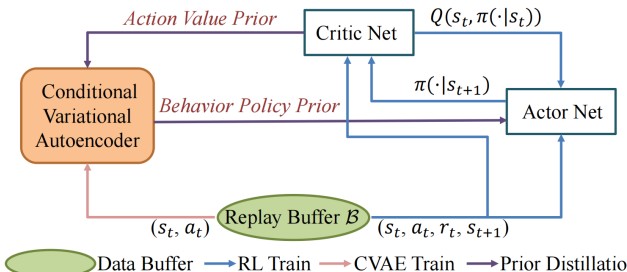

Figure 1: We introduce prior knowledge distillation to enhance learning efficiency and propose bidirectional distillation mechanisms: integrating action value priors into the conditional variational autoencoder (CVAE) (Kingma et al., 2014) network training, and incorporating behavior policy priors during policy updates.

minimal modifications. The contributions of this work are threefold. First, we theoretically show that the generative support-set behavior prior is sufficient to improve exploration efficiency by producing high-value policy priors. Building on this insight, we propose a Bidirectional Behavior Prior Distillation (B2PD) algorithm that generates policy priors proactively to guide the policy update process in online RL. Second, we provide a theoretical analysis of the regularization effect of action noise and introduce a noise scheduling mechanism that enables stable Q-value estimation, thereby filtering high-quality update anchors from the policy support set and facilitating stable and efficient policy improvement. Third, through qualitative toy experiments, we show that B2PD exhibits stronger exploration capabilities and yields higher-quality policies. Extensive evaluations on both state-based and pixel-based tasks further demonstrate its superior sample efficiency and enhanced training stability.

## 2 PRELIMINARIES

### 2.1 REINFORCEMENT LEARNING

Within the standard framework of the Markov decision process (MDP), RL can be formulated as $\mathcal{M} = \langle \mathcal{S}, \mathcal{A}, \mathcal{P}, \mathcal{R}, \gamma \rangle$. Here, $\mathcal{S}$ denotes the state space, $\mathcal{A}$ denotes the action space, $r : \mathcal{S} \times \mathcal{A} \in [-\mathcal{R}_{max}, \mathcal{R}_{max}]$ denotes the reward function, and $\gamma \in (0, 1)$ is the discount factor, and $\mathcal{P}(\cdot \mid s, a)$ stands for transition dynamics. For simplicity, we denote the current and next state-action pairs as $(s_t, a_t)$ and $(s_{t+1}, a_{t+1})$, respectively. The agent's behavior is defined by a stochastic policy $\pi(\cdot|s_t)$, which maps a given state to a probability distribution over possible actions. The state and state-action distributions induced by $\pi$ are denoted by $\rho_\pi(s_t)$ and $\rho_\pi(s_t, a_t)$, respectively.

## 2.2 MAXIMUM ENTROPY RL

In our work, we adopt an entropy-augmented objective (Haarnoja et al., 2017), which incorporates policy entropy into the reward signal to define the policy optimization objective as

$$J_{\text{MaxEnt}}(\phi) = \sum_{t=0}^{T} \mathbb{E}_{(s_t,a_t) \sim \rho_\pi} \Big[ r(s_t, a_t) + \alpha \mathcal{H}\left(\pi(\cdot|s_t)\right) \Big], \tag{1}$$

where $\alpha$ is the temperature coefficient, and the policy entropy $\mathcal{H}$ is expressed as

$$\mathcal{H}(\pi(\cdot|s_t)) = \mathbb{E}_{a_t \sim \pi_\phi} \big[ -\log \pi(a_t|s_t) \big]. \tag{2}$$

The optimal policy can be obtained via a maximum entropy variant of policy iteration, which comprises two alternating steps: (a) soft policy evaluation and (b) soft policy improvement. This procedure is collectively referred to as soft policy iteration. Given a policy $\pi$, the corresponding soft Q-value can be learned by iteratively applying a modified Bellman operator $\mathcal{T}_c^\pi$ that incorporates the entropy term $\mathcal{H}$, which can be formulated as

$$\mathcal{T}_c^\pi Q(s_t, a_t) = r(s_t, a_t) + \gamma \mathbb{E}_{s_{t+1} \sim \rho_\pi} \big[ \mathbb{E}_{a_{t+1} \sim \pi_\phi} [Q_{\theta'}(s_{t+1}, a_{t+1}) - \alpha \log \pi_\phi(\cdot|s_{t+1})] \big], \tag{3}$$

where $\gamma \in [0, 1)$ is the discount factor, $\theta$ denotes the parameters of the Q-network, and $\phi$ denotes the parameters of the Gaussian policy. The parameters of the soft Q-function are optimized by minimizing the soft Bellman residual, which can be formalized as

$$\mathcal{L}_Q(\theta) = \mathbb{E}_{(s_t,a_t) \sim \mathcal{B}}(Q(s_t, a_t) - \mathcal{T}_c^\pi Q(s_t, a_t))^2, \tag{4}$$

where $\mathcal{B}$ denotes a mini-batch sampled from the replay buffer.

## 3 METHODS

### 3.1 POLICY DISTILLATION

Specifically, under the assumptions of Universal Multimodal Policy Approximation (Huang et al., 2023) and Policy Support Set Inclusion (Sohn et al., 2015; Wang et al., 2025), Proposition 3.3 establishes the existence of a high-quality policy prior. This result ensures that the support set of the behavioral policy alone is sufficient to induce a superior policy prior, namely, $Q_\theta(s_t, \tilde{a}) \geq Q_\theta(s_t, \pi_\phi(s))$.

**Assumption 3.1** (Universal Multimodal Policy Approximation (Huang et al., 2023)). Let CVAE $G_\omega$ denote the class of behavior policies parameterized by $\omega$, where the latent variable $z \sim p(\cdot|s_t)$ captures context-dependent stochasticity and the decoder $q(\cdot|s_t, z)$ maps to actions. Then, under sufficient model capacity and data coverage, for any continuous and potentially multimodal optimal policy, there exists $G_\omega(\cdot|s_t)$ that approximates the actor $\pi_\phi(\cdot|s_t)$.

Moreover, due to its ability to effectively model complex multidimensional multimodal distributions, the expressive power of the behavior policy network $G$ strictly exceeds that of the unimodal Gaussian policies commonly employed in maximum entropy RL.

**Assumption 3.2** (Policy Support Set Inclusion (Sohn et al., 2015; Wang et al., 2025)). Let $\Pi_{G_\omega}$ denote the set of actions with non-zero density under a CVAE-based policy support set $\Pi_{G_\omega(\cdot|s_t)}$, and $\pi_\phi(\cdot|s_t)$ denote a Gaussian policy. Then, for any $s_t \in \mathcal{S}$, $\pi_\phi(\cdot|s_t) \subsetneq \Pi_{G_\omega(\cdot|s_t)}$, where the inclusion is strict if $\Pi_{G_\omega}$ approximates a multimodal policy distribution.

Based on these two assumptions, we propose the following proposition to characterize the conditions under which expert prior actions exist, enabling sampling from the generative model and distilling prior knowledge into the agent.

**Proposition 3.3.** *Suppose $s$, $a$, and $r$ follow MDP model. Under the assumption that the generative model is universal policy approximation and multimodal, there exists at least one action $\tilde{a} \in \Pi_{G_\omega}(\cdot|s_t)$ such that $Q_\theta(s_t, \tilde{a}) \geq Q_\theta(s_t, \pi_\phi(\cdot|s_t))$.*

Note that this proposition guarantees the superiority of the generative policy in an asymptotic manner. Furthermore, by imposing additional assumptions on the policy generation mechanism, we can uniquely determine the existence of the expert policy prior within the distribution $\Pi_{G_\omega}$. For a complete discussion of the proof, please refer to the proof in Appendix B.1.

## 3.2 PRACTICAL ALGORITHM

**Standard Deviation-Aware Noise Scheduling.** We begin by training the Q-network to provide action-value estimation prior knowledge for the CVAE. To achieve stable Q-value estimation, we propose the SDA Noise scheduling mechanism, which dynamically adjusts the noise magnitude based on the predicted action distribution. This further refines the Q-target value estimation process in the vanilla SAC (Haarnoja et al., 2018) algorithm.

**Theorem 3.4** (Noise-Regularized Q-Value Smoothness)**.** *The soft Q-value $Q(s_t, a_t)$ is assumed to be Lipschitz continuous (Geist et al., 2019). Moreover, the Q-function $Q(s_t, a_t)$ is twice differentiable concerning $a$, and there exists two constants $C_1 > 0$, $C_2 > 0$ such that $\|\nabla_a Q(s_t, \hat{a})\| \leq C_1, \|\nabla_a^2 Q(s_t, \hat{a})\| \leq C_2, \quad \forall \hat{a} \in (a_t, a_t + \epsilon)$, That is, for all states $s_t \in \mathcal{S}$ and Gaussian action noise $\epsilon \sim \mathcal{N}(0, \tau^2 \mathbf{I})$, the following inequality holds*

$$\mathbb{E}_\epsilon |Q(s_t, a_t) - Q(s_t, a_t + \epsilon)| \leq C_1 \sqrt{\frac{2}{\pi}} \tau + \frac{1}{2} C_2 \tau^2, \tag{5}$$

*where $\tau$ denotes the standard deviation of the Gaussian noise. This smoothing effect facilitates more stable Q-value estimation and mitigates high variance in bootstrapped Q-learning. The proof is presented in Appendix B.1.*

Under the maximum entropy framework, adding Gaussian noise $\epsilon$ to actions yields smoother and more stable action-value estimates, formulated as

$$\mathcal{T}_c^\pi Q_\theta(s_t, a_t) = r + \gamma \mathbb{E}_{s_{t+1} \sim \rho_\pi, a_{t+1} \sim \pi_\phi} \left[ Q_{\theta'}(s_{t+1}, a_{t+1} + \epsilon) - \alpha \log \pi_\phi(a_{t+1} \mid s_{t+1}) \right]. \tag{6}$$

In the SAC algorithm, action noise may lead to abrupt changes in entropy, as it ignores the impact of channel-wise noise variations on the policy entropy. Therefore, we exclude the noise-perturbed actions from the entropy computation.

To achieve stable Q-value estimation, we propose an SDA Noise scheduling mechanism. Specifically, the action distribution is modeled as a Gaussian policy by the actor network, defined as

$$\pi_\phi(\cdot|s_t) = \mathcal{N}\left(\mu_\phi(s_t), \mathrm{diag}(\sigma_\phi^2(s_t))\right), \tag{7}$$

where $\mu_\phi(s_t)$ denotes the mean and $\mathrm{diag}(\sigma_\phi^2(s_t))$ represents the diagonal covariance matrix parameterized by the predicted standard deviation vector $\sigma_\phi(s_t)$. We design a noise scheduling function that adaptively scales the action noise, where the noise standard deviation $\tau_t$ is formulated as

$$\tau_t = \frac{1}{d_a} * \sum_{i=1}^{d_a} \frac{0.2}{\exp\left(\sigma_\phi^i(s_t)\right)^{1.5}}, \tag{8}$$

where $d_a$ denotes the dimensionality of the action space, and $\sigma_t^i$ is the predicted standard deviation vector of the $i$-th element at time step $t$. This formulation is closely related to the noise injection mechanism in TD3 and adopts the same maximum noise magnitude of 0.2. This mechanism allows the agent to adaptively adjust the noise standard deviation based on the uncertainty of the predicted action distribution. When the predicted action standard deviation is large, the noise standard deviation $\tau_t$ correspondingly decreases, resulting in a reduced exploration range that benefits the stability of Q-value estimation. Conversely, a decrease in the action standard deviation expands the exploration range, facilitating broader exploration of the action space.

**Action Value Prior Distillation.** In this section, we provide a comprehensive overview of the construction of the behavior policy model. Considering the mode collapse issue commonly observed in GANs (Liu et al., 2019), we avoid using GAN-based architectures to learn the policy prior. Consequently, we adopt a CVAE to build the behavior policy model $G$. During action reconstruction, the model incorporates action value estimates as prior knowledge to guide $G$ toward producing higher-value actions. The hyperparameter configuration of $G$ is detailed in Appendix C.2.

The CVAE $G_\omega$, parameterized by $\omega$, consists of an encoder $p_\omega(s_t, a_t)$ and a decoder $q_\omega(s_t, z)$, denoted as $G = \{p, q\}$. The CVAE is optimized by maximizing its evidence lower bound objective (ELBO), which can be formulated as

$$\mathcal{V}_{\mathrm{ELBO}}(\omega) = \mathbb{E}_{(s_t, a_t) \sim \mathcal{B}, z \sim p_\omega(s_t, a_t)} \left[ (a_t - q_\omega(s_t, z))^2 + \mathrm{KL}\left(p_\omega(s_t, a_t) || \mathcal{N}(0, \mathbf{I})\right) \right], \tag{9}$$

where $\mathrm{KL}(p||q)$ denotes the kullback–leibler (KL) (Hinton et al., 2015) divergence between the probability distributions $p(\cdot)$ and $q(\cdot)$, and $\mathbf{I}$ represents the identity matrix. When sampling actions from the CVAE, we first sample a latent variable $z$ from the prior distribution, which is assumed to follow a normal distribution $\mathcal{N}(0, \mathbf{I})$. This latent variable, along with the state $s_t$, is then passed into the decoder $q_\omega(s_t, z)$ to obtain the decoded action.

Furthermore, we expect the value estimation prior in the Q-network to serve as an effective guidance signal, enabling the CVAE to generate actions with higher expected returns. As noted by DIDI (Liu et al., 2024), employing Q-value guided gradients to train a generative model yields diverse and high-value actions. We propose an action-value prior distillation loss, which leverages double Q-networks to guide the gradient update direction of the CVAE. The Q-value prior distillation loss can be formulated as

$$\mathcal{V}_{\mathrm{Dist}}(\omega) = -\mathbb{E}_{(s_t, a_t) \sim \mathcal{B}, z \sim p_\omega(s_t, a_t)} Q_\theta(s_t, q_\omega(s_t, z)). \tag{10}$$

We optimize the complete CVAE model by integrating the two aforementioned loss components. To this end, we introduce a weighting parameter $\xi$ to balance the trade-off between action reconstruction and the value of the generated actions. The overall loss function is then formulated as

$$\mathcal{V}_G(\omega) = \mathcal{V}_{\mathrm{ELBO}}(\omega) + \xi \mathcal{V}_{\mathrm{Dist}}(\omega). \tag{11}$$

While both the CVAE and the actor network rely on the Q-network to optimize action selection, Theorem 3.3 shows that the CVAE enjoys a substantially stronger fitting capacity than the unimodal Gaussian actor head. As a result, the action support set sampled from the CVAE can more effectively guide actor optimization and suppress inefficient exploration. This insight naturally leads to our Behavior Policy Prior Distillation module.

**Behavior Policy Prior Distillation.** We leverage the behavior policy model $G$ to generate the policy prior $G_\omega(s)$, which provides a high-value anchor for policy updates during the actor optimization. The behavior prior is obtained via multiple forward passes of the generative network derived from the policy support set, where the anchor $\tilde{a}$ for policy updates is chosen based on Q-target evaluations, and is formulated as

$$\tilde{a} = \arg\max_{a_h} \left( Q_{\theta'}(s_t, a_h) \right), a_h \in \Pi_{G_\omega(\cdot|s_t)}, \tag{12}$$

where $\arg\max_{a_h}$ denotes the process that identifies optimal actions from the policy prior set $\Pi_{G_\omega(\cdot|s_t)}$, and $h = [1, 2, 3, \dots, H]$ denotes the indices of $H$ prior actions generated by the CVAE. Following the identification of the optimal action, we implement a systematic optimization mechanism that progressively refines the policy network with the optimal actions through a knowledge distillation paradigm consistent with the algorithm established in (Guo et al., 2024). We incorporate expert prior knowledge into the actor network by minimizing the KL divergence as the training objective, which can be formalized as

$$J_{\mathrm{Dist}}(\phi) = -\mathbb{E}_{s_t \sim \mathcal{B}} \left[ \mathrm{KL}(\pi_\phi(\cdot|s_t)||\tilde{a}) \right]. \tag{13}$$

Then, we integrate the maximum entropy objective with the expert policy prior distillation loss. The overall loss function of the Actor network can be formulated as

$$J_\pi(\phi) = J_{\mathrm{MaxEnt}}(\phi) + \eta J_{\mathrm{Dist}}(\phi), \tag{14}$$

where $\eta$ denotes the weight of the policy prior distillation loss $J_{\mathrm{Dist}}$.

### 3.3 A Toy Example

To provide a more intuitive view of the effectiveness of B2PD compared to maximum entropy reinforcement learning methods, we conduct a comparative analysis of trajectories generated by B2PD and SAC at different training steps. We focus on two evaluation aspects: (1) Can B2PD reduce inefficient random exploration? (2) Does the trajectory distribution cover high-reward regions?

We construct a toy environment with a state space formed by a mixture of eight Gaussian reward centers, where $S = [S_x, S_y] = [0, 6]^2$ denote the state dimensions, the action space is defined as $A = [A_x, A_y] = [-1, 1]^2$, and the state transition follows $s_{t+1} = s_t + a_t \times 0.1$. To examine whether the agent exhibits inefficient exploration during initial training, we estimate the kernel density of

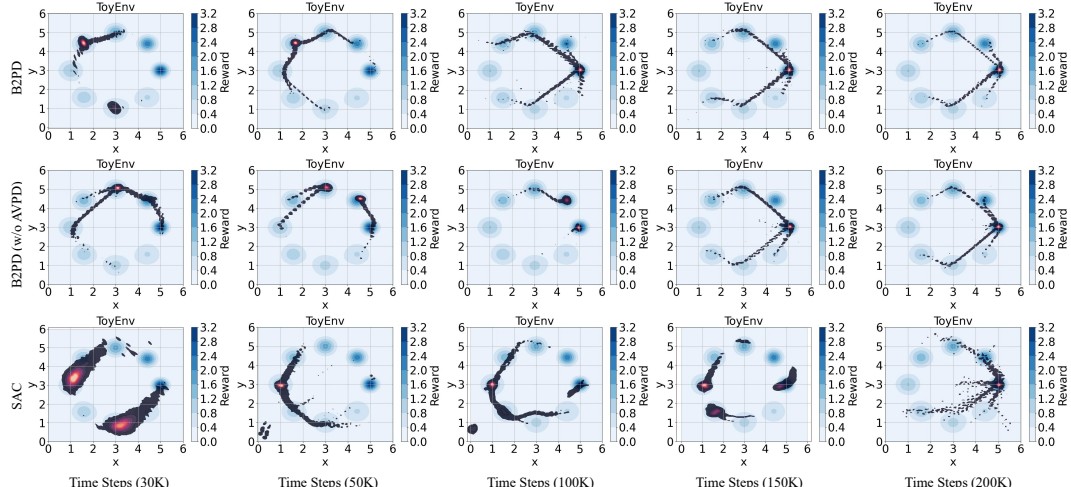

Figure 2: Visualization of reward and state visitation in the ToyEnv. Blue shading indicates the reward magnitude, with darker blue representing higher rewards. Red shading corresponds to the state visitation frequency, where more intense red reflects states visited more frequently. Regions shown in white or pure blue indicate areas that are rarely or never explored. Each row corresponds to one of the evaluated algorithms (B2PD, B2PD (w/o AVPD), and SAC), and each column corresponds to different training timesteps (30K, 50K, 100K, 150K, and 200K).

state distributions from validation trajectories and visualize the visitation frequency across the state space. We visualize the state trajectory distributions at training timesteps 30K, 50K, 100K, 150K, and 200K, using 100 sampled trajectories per checkpoint, as illustrated in Figure 2.

From these visualizations, we draw three key observations: First, the results show that B2PD tends to explore nearby high-reward regions progressively. This structured exploration strategy effectively reduces inefficient exploration. In contrast, SAC often falls into local optima even at 150K timesteps, indicating that blind exploration driven by stochastic policy and entropy regularization leads to suboptimal learning efficiency. Second, compared with SAC, B2PD significantly reduces the visitation frequency of low-reward states, exhibiting superior sample efficiency. We attribute this to B2PD explicitly distilling the expert prior policy into the agent, such that each state transition is guided by expert action anchors, thereby reducing ineffective exploration. Third, compared with B2PD (w/o AVPD), the policy validation trajectory of B2PD with action-value priors converges noticeably faster. This improvement arises because the policy updates are more effectively guided toward value-driven high-reward regions, enabling accelerated convergence.

We argue that the entropy-driven exploration mechanism in SAC expands the action distribution in a largely uniform manner, encouraging broad exploration but ignoring the underlying value structure. Consequently, SAC may spend substantial effort exploring inefficient or redundant regions of the state–action space, particularly when high-reward areas constitute only a small subset, as shown in Figure 2. In contrast, B2PD restricts exploration toward value-aligned, high-quality modes encoded in the distilled prior. This induces a more structured exploration pattern, where the policy deliberately avoids low-value regions that contradict the prior knowledge, ultimately leading to more efficient exploration and faster convergence. A complete comparison is provided in Appendix D.

## 4 EXPERIMENTS

In this section, we evaluate the performance of B2PD using two types of observations. For state-based inputs, we consider seven MuJoCo continuous-control tasks, four PyBullet tasks, and four DMControl suite tasks. For visual inputs, we evaluate the method on four pixel-based DMControl tasks. These experiments aim to answer the following research questions: (1) Can B2PD leverage behavior policy to accelerate online learning? (2) Why does behavior prior to distillation outperform entropy-driven exploration? (3) Does the algorithmic module built upon environment-specific design choices yield consistently reliable performance? For (1), we first qualitatively demonstrate

the effectiveness of B2PD through the toy experiments in Section 3.3. We then provide a quantitative comparison between B2PD and existing methods that leverage policy priors to accelerate online learning. For (2), we analyze the differences between policy prior distillation and the original algorithm in a toy experiment. Lastly, for (3), we conduct experiments using the B2PD algorithm with a fixed set of parameters across three challenging environments.

## 4.1 EXPERIMENTAL SETUP

**Baselines.** To demonstrate the effectiveness of our proposed method, we compare it against representative baselines, including deterministic and stochastic policy gradients (DDPG (Lillicrap, 2015), TD3 (Fujimoto et al., 2018), SAC (Haarnoja et al., 2018)), augmented state representations (ALH (Quang & Lauw, 2024a)), and policy distillation with nearest-neighbor guidance (NNPG (Shen & Yang, 2021)). For a fair comparison with our proposed method, we provide a detailed specification of the parameters used in the baselines (see Appendix C.2).

(1) **DDPG (Lillicrap, 2015)**: DDPG addresses the optimization challenge in continuous action spaces through the Actor-Critic framework and deterministic policy gradients, while enhancing stability via experience replay and target networks.

(2) **TD3 (Fujimoto et al., 2018)**: TD3 is a prominent off-policy deterministic policy gradient algorithm that stabilizes training by incorporating a double Q-network, delayed policy updates, and target policy smoothing regularization.

(3) **SAC (Haarnoja et al., 2018)**: SAC is a stochastic policy algorithm built upon the maximum entropy reinforcement learning framework, which achieves efficient learning in continuous action spaces by maximizing a trade-off between expected return and policy entropy.

(4) **ALH (Quang & Lauw, 2024a)**: ALH is an improvement built upon the TD3 algorithm, serving as a decision enhancement method that leverages state representations from the previous timestep to improve the perceptual capability of the Actor network and enhance the decision-making process.

(5) **Nearest Neighbor Policy Guidance (NNPG)**: We implement the nearest-neighbor exploration algorithm proposed in NNAC (Shen & Yang, 2021) based on the SAC algorithm to address the challenge of efficient exploration in continuous action spaces. NNPG leverages the nearest neighbor action of the current policy decision as an anchor to guide the actor update.

(6) **B2PD (TD3)**: Built upon the TD3, the B2PD algorithm additionally trains a CVAE network, which provides expert behavior prior for policy improvement by leveraging action-value prior distillation. Subsequently, B2PD leverages an expert prior distillation loss to guide the Actor update, thereby enabling more stable and efficient policy updates for the RL agent.

(7) **B2PD**: Building upon the SAC framework, we establish a bidirectional knowledge transfer between the agent and the behavior policy network ($G_\omega$), enabling the distillation of expert behavior priors into the agent. The NNPG and B2PD algorithms incorporate policy distillation techniques to refine policy updates, aiming to enhance sample efficiency and reduce ineffective policy exploration.

For pixel-based environments, we adopt RAD (Laskin et al., 2020) and DrQ-v2 (Yarats et al., 2022) as baselines and integrate our method with both frameworks to assess its effectiveness under visual observations. To ensure architectural consistency, the CVAE used in B2PD-RAD and B2PD-DrQ-v2 shares the same CNN backbone as the encoder $p_\omega$, while only introducing an additional decoder $q_\omega$ to generate the policy prior.

**Benchmark environments.** We conduct comprehensive evaluations of the proposed method across 15 tasks from the MuJoCo, PyBullet, and DMControl environments, as well as 4 visual control tasks from DMControl environments. Detailed descriptions of these control environments are provided in Appendix C.

**Hyperparameters.** In Section 3, we introduce three parameters: an action value prior coefficient $\xi$=0.1, the number of sampled actions $H$=10, and a behavior prior coefficient $\eta$=1.0. Unless otherwise stated, we maintain consistent hyperparameter configurations across all experiments. Regarding model architecture, our method strictly replicates the structural framework of the SAC algorithm. All experiments are conducted in a Linux environment equipped with a 56-core Xeon(R) 6133 CPU

and 1 Nvidia RTX 3090 GPU. We further conduct a sensitivity analysis on these hyperparameters (see Appendix F).

**Random seeds.** To ensure the reproducibility of B2PD, we evaluated each algorithm using 10 random seeds. Furthermore, we maintained consistent seeds across all experiments, applying them to PyTorch, Numpy, Gym, and CUDA packages.

**Evaluation.** No data or parameters were reused during evaluation, and each evaluation step consisted of 10 randomly initialized episodes. We train for 2 million timesteps on MuJoCo (Todorov et al., 2012) and PyBullet (Coumans & Bai, 2016) environments, while only 500K timesteps are used for training on the DMControl (Tassa et al., 2018) environments. The policy is evaluated every 5000 timesteps. We present the mean and 95% confidence interval over the final 10 trials.

## 4.2 EVALUATION ON VARIOUS BENCHMARK SUITES

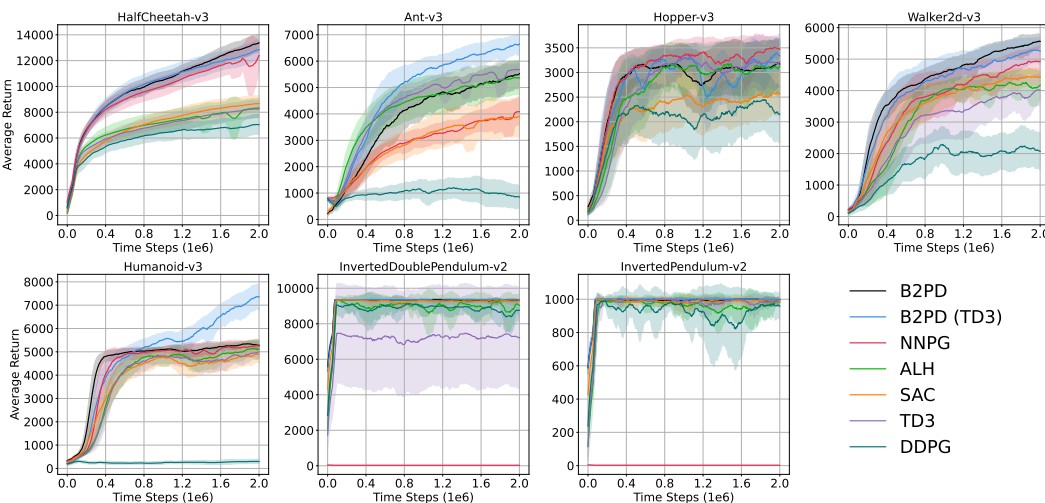

Figure 3: Learning curves on seven MuJoCo continuous control tasks. The shaded area captures a 95% confidence interval around the average performance over 10 trials. Curves are smoothed uniformly for visual clarity.

| Tasks | DDPG | TD3 | SAC | ALH | NNPG | B2PD (TD3) | B2PD |
|---|---|---|---|---|---|---|---|
| HalfCheetah-v3 | 7030±772 | 8259±649 | 8672±602 | 8339±730 | 12429±1048 | 12877±778 | **13386±694** |
| Ant-v3 | 848±431 | 5697±202 | 3906±766 | 5361±644 | 4083±523 | **6632±398** | 5536±446 |
| Hopper-v3 | 2150±578 | 3216±561 | 2552±483 | 3130±554 | **3467±253** | 3321±351 | 3175±489 |
| Walker2d-v3 | 2045±587 | 4008±485 | 4419±259 | 4159±457 | 4978±439 | 5247±427 | **5568±220** |
| Humanoid-v3 | 297±90 | 5003±255 | 4892±346 | 5117±422 | 5243±274 | **7359±573** | 5276±183 |
| InvertedDouble-v2 | 8766±580 | 7293±2829 | 9293±82 | 9088±542 | 28±1 | **9358±4** | 9344±20 |
| InvertedPendulum-v2 | 954±90 | 995±9 | 991±16 | 985±29 | 2±0 | **1000±0** | 996± 9 |

Table 1: Average return of the last 10 evaluation scores on MuJoCo environments over 10 random seeds, where ± captures a 95% confidence interval. The maximum values of each row are bolded. "InvertedDouble-v2" denotes the InvertedDoublePendulum-v2 task for brevity.

**Main results on MuJoCo environments.** We compare our proposed B2PD algorithm to these baselines, and these comparisons are performed on seven MuJoCo continuous control tasks. Learning curves are displayed in Figure 3, and final results are listed in Table 1.

Figure 3 shows that the proposed B2PD and B2PD (TD3) algorithms consistently outperform vanilla algorithms across multiple tasks, demonstrating significant performance gains and more stable training compared to other policy-prior guidance approaches (NNPG, ALH). Specifically, on the benchmark tasks InvertedDouble (denoted as InvertedDoublePendulum) and InvertedPendulum, the

NNPG fails to learn effective control policies. We attribute this to the lack of diverse policy anchors as optimization references during the early training phase, which results in overly conservative policy anchors being provided for relatively easy tasks (with action dimensions of 1 and 1, respectively). In contrast, the B2PD algorithm significantly improves performance by generating diverse, high-value policy priors to guide the policy update process.

As shown in Table 1, our proposed B2PD method achieves state-of-the-art (SOTA) performance on 6 out of 7 MuJoCo control tasks. At 2M timesteps, B2PD yields an average performance improvement of 22.2% over the vanilla SAC algorithm, while B2PD (TD3) surpasses the vanilla TD3 by 26.1%. Although the ALH incorporates historical state representations and employs more complex temporal architectures, B2PD consistently demonstrates stable and significant advantages across most tasks, further validating the generality and effectiveness of our method.

**Main results on state and pixel-based DMControl environments.** We evaluate the proposed B2PD algorithm on state-based DMControl benchmarks using both TD3 and SAC as underlying actor–critic learners. To further assess its generality in visual-input scenarios, we integrate B2PD with RAD and DrQ-v2, and report the corresponding learning curves in Figure 4 and final performance comparisons in Table 2. Experimental results demonstrate that the proposed B2PD algorithm significantly accelerates policy convergence. Across all four tasks, both B2PD and B2PD (TD3) consistently outperform the vanilla SAC and TD3 algorithms, respectively, further demonstrating the effectiveness and generality of our method.

| Settings | state | | | | pixel | | | |
|---|---|---|---|---|---|---|---|---|
| Tasks | TD3 | SAC | B2PD (TD3) | B2PD | RAD | DrQ-v2 | B2PD-RAD | B2PD-DrQ-v2 |
| cartpole-swingup-v1 | 766±23 | 695±39 | **869±5** | 858±3 | 595±191 | 758±173 | 487±234 | **828±36** |
| reacher-easy-v1 | 963±13 | 946±29 | 979±6 | **980±5** | 579±173 | 738±121 | 777±110 | **938±38** |
| cheetah-run-v1 | 490±43 | 482±32 | **782±39** | 764±43 | 202±88 | **566±127** | 400±327 | 525±47 |
| walker-walk-v1 | 834±89 | 835±85 | **939±39** | 863±90 | 631±136 | 605±290 | **856±47** | 840±29 |

Table 2: Average return of the last 10 evaluation scores on DMControl environments over 10 random seeds, where ± captures a 95% confidence interval. The maximum values of each row are bolded.

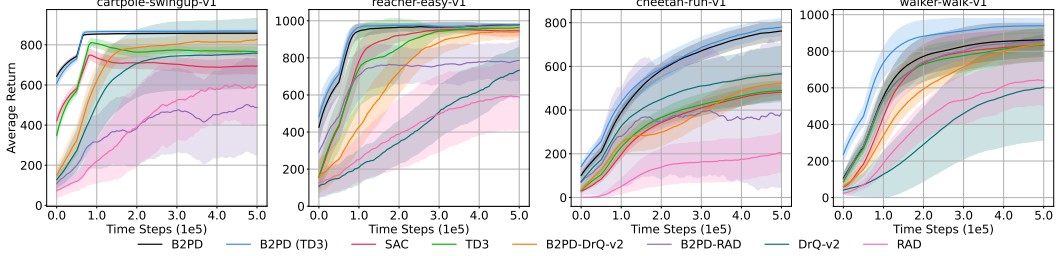

Figure 4: Learning curves on four DMControl tasks. The shaded area captures a 95% confidence interval around the average performance over 10 trials. Curves are smoothed uniformly for visual clarity.

The performance of the pixel-based RAD and DrQ-v2 is inferior to that of state-based methods on tasks such as reacher-easy, cheetah-run, and walker-walk. However, when DrQ-v2 is combined with B2PD, its performance exceeds that of state-based methods on cartpole-swingup and walker-walk, and it achieves comparable performance on reacher-easy. This further demonstrates that the B2PD method is effective on both state- and pixel-based tasks, highlighting its robustness across observation modalities.

### 4.3 ABLATION EXPERIMENT

**Effects of each mechanism.** We conduct ablation studies on the Ant task to evaluate the individual contributions of each component in our method, including the SDA Noise scheduling and the

AVPD modules. The results are presented in Figure 5. The experiments demonstrate that each component is essential to the overall performance of B2PD. Notably, B2PD maintains a high-entropy action distribution similar to SAC during initial training but achieves rapid policy convergence and low-entropy behavior in later stages, highlighting the role of policy distillation in stabilizing optimization. This demonstrates that B2PD effectively leverages policy priors to guide updates and reduce inefficient exploration. Furthermore, as our method relies on accurate value estimation from the Q-network, the SDA Noise scheduling mechanism serves as an essential part of the proposed algorithm. We provide a comprehensive and systematic comparison of these two configurations in the MuJoCo environments in the Appendix E, which highlights the significant impact of distillation loss selection on policy effectiveness.

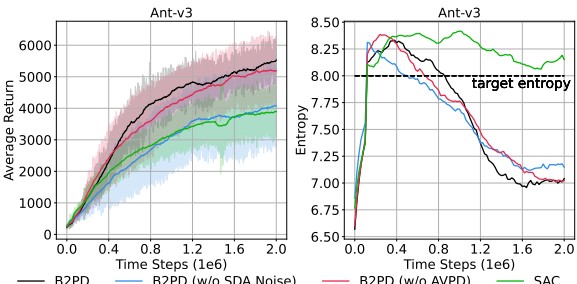

Figure 5: Ablation study on B2PD components. The results show that its mechanisms collaboratively improve performance and reduce inefficient exploration. The shaded area captures a 95% confidence interval around the average performance over 10 trials. Curves are smoothed uniformly for visual clarity.

## 5 CONCLUSIONS AND LIMITATIONS

This work investigates the potential benefits of incorporating various forms of policy priors into the policy learning process. Building on this insight, we theoretically show that employing a sufficiently expressive distribution approximator can provide policy prior guidance to a Gaussian actor network during policy updates. Motivated by this observation, we propose B2PD to address the challenge of inefficient exploration. Through extensive empirical evaluation, we demonstrate that, under fixed hyperparameter settings, B2PD improves algorithmic performance on both state-based and pixel-based tasks. Qualitative analyses in toy environments further show that B2PD effectively suppresses exploration in low-value regions by distilling high-value policy priors into the agent. Collectively, these findings indicate that policy distillation can mitigate the ineffective exploration caused by high-entropy policy training. We hope that this work advances online RL toward low-entropy exploration and provides new insights for the field.

Despite demonstrating significant performance improvements in experiments, B2PD still exhibits certain notable limitations. Specifically, as discussed in Appendix D, B2PD achieves rapid convergence in toy experiments with dense rewards; however, since the action value prior heavily relies on Q-value estimation, further investigation is needed to obtain high-value priors in sparse reward tasks.

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

# A RELATED WORKS

## A.1 ONLINE RL WITH STATIC POLICY PRIOR

In contrast to training agents directly from scratch, RL with static policy prior (Walke et al., 2023; Ball et al., 2023; Zang et al., 2023; Spigler, 2024; Choi & Seo, 2025) leverages explicit knowledge obtained from offline datasets to guide online agent training, thereby accelerating policy convergence. In practical applications, these additional priors are typically used to direct the agent's focus on high-value areas of the Markov Decision Process (MDP), reducing unnecessary exploration.

Initially, these data are employed to initialize policies through behavior cloning (BC) (George & Farimani, 2023), and they permeate throughout the optimization process. For instance, in RLPD (Ball et al., 2023) and EXPLORE (Li et al., 2023), demonstration data are incorporated into the replay buffer, and new regularization terms are introduced to force the optimization process to better utilize the demonstration data. Similarly, POfD (Kang et al., 2018), LOGO (Rengarajan et al., 2022), BPR (Zang et al., 2023), PPD (Spigler, 2024), and DCSL (Choi & Seo, 2025) impose penalties or constraints on the RL objective, forcing the agent's policy to remain close to the policy prior. However, these methods are only feasible when expert policies are available, as they cannot effectively guide policy updates in states with low-quality interactions (Hao et al., 2023; Xudong et al., 2024b;a; Beliaev & Pedarsani, 2025). Furthermore, these approaches might require substantial data to direct the RL agent, and acquiring such data can be challenging in systems with high sampling costs.

## A.2 ONLINE RL WITH DYNAMIC POLICY PRIOR

Dynamic policy prior offers a more flexible framework that does not rely on offline data. Existing online RL algorithms with dynamic policy priors typically leverage local policy guidance to facilitate agent exploration, either by integrating historical trajectory data to enrich policy representation (Kapturowski et al., 2018; Quang & Lauw, 2024b), or by constraining the exploration space to improve learning efficiency (Shen & Yang, 2021; Goyal et al., 2022; Zhao et al., 2023; Guo et al., 2024; Daoudi et al., 2024).

On the one hand, incorporating historical interactions is an intuitive strategy to enhance the decision-making process. For example, R2D2 (Kapturowski et al., 2018) leverages stored LSTM (Yu et al., 2019) hidden states for initialization, employs a 50-step burn-in period, and performs a 10-step history rollout during training, thereby utilizing recurrent representations to capture long-term temporal dependencies. Similarly, ALH (Quang & Lauw, 2024b) improves decision-making by using past observation states to guide the current agent's actions, leading to notable performance gains. However, due to cognitive uncertainty in the early stages of RL training, the accumulation of errors may result in suboptimal performance.

On the other hand, nearest-neighbor RL (Shah & Xie, 2018) algorithms implement constrained policy exploration and reduce ineffective exploration due to Q-value estimation bootstrapping. MSR (Gong et al., 2025) introduces a robust solution to the policy discontinuity problem by incorporating the policy discrepancy between adjacent goals as a regularization term and constraining it within a minimally acceptable threshold, thereby preventing abrupt changes in actions. RLLG (Daoudi et al., 2024) proposed an RL algorithm based on noisy policy switching, which effectively leverages local guide policies to significantly enhance the performance of an approximate policy evaluation (APE)-based RL algorithm, with particularly notable improvements during the early stages of training. Our proposed method shares similarities with prior works (Shen & Yang, 2021; Daoudi et al., 2024), as it also leverages a local expert guide as policy prior knowledge to inform and regularize the updates of the policy network. The core of our method lies in leveraging the CVAE to generate higher-quality actions, which incorporates action-value prior distillation during the policy network update and investigates how policy prior distillation mitigates the ineffective exploration inherent in maximum entropy RL.

## A.3 CONNECTION WITH OFFLINE-TO-ONLINE REINFORCEMENT LEARNING

We highlight the interconnections between offline RL (Ernst et al., 2005; Lyu et al., 2022; Levine et al., 2020) and the transition from offline-to-online settings. A significant portion of prior works have involved performing offline RL followed by online fine-tuning (Nakamoto et al., 2023; Nair

et al., 2020; Ball et al., 2023; Wagenmaker & Pacchiano, 2023). The core of our method lies in incorporating action-value prior distillation into the behavior prior network, thereby leveraging the CVAE to generate higher-quality actions. This approach alleviates the inherent inefficiency of exploration in maximum entropy RL, enabling more efficient policy improvement. Compared to Offline-to-Online RL, our proposed method introduces a generative policy prior to guide agent updates, thereby enhancing online learning performance without requiring offline pretraining and significantly reducing system complexity.

# B THEORETICAL ANALYSES

## B.1 BEHAVIOR PRIOR DISTILLATION

We begin by providing the definitions of the approximation assumption and the support coverage assumption, which are utilized in our theoretical analysis.

**Assumption B.1** (Universal Multimodal Policy Approximation (Huang et al., 2023)). Let CVAE $G_\omega$ denote the class of behavior policies parameterized by $\omega$, where the latent variable $z \sim p(\cdot|s_t)$ captures context-dependent stochasticity and the decoder $q(\cdot|s_t, z)$ maps to actions. Then, under sufficient model capacity and data coverage, for any continuous and potentially multimodal optimal policy, there exists $G_\omega(\cdot|s_t)$ that approximates the actor $\pi_\phi(\cdot|s_t)$.

**Assumption B.2** (Policy Support Set Inclusion (Achiam et al., 2017)). Let $\Pi_{G_\omega}$ denote the set of actions with non-zero density under a CVAE-based policy support set $\Pi_{G_\omega(\cdot|s_t)}$, and $\pi_\phi(\cdot|s_t)$ denote a Gaussian policy. Then, for any $s_t \in \mathcal{S}$, $\pi_\phi(\cdot|s_t) \subsetneqq \Pi_{G_\omega(\cdot|s_t)}$, where the inclusion is strict if $\Pi_{G_\omega}$ approximates a multimodal policy distribution.

**Proposition B.3.** *Suppose $s_t$, $a_t$, and $r_t$ follow MDP model. Under the assumption that the generative model is universal policy approximation and multimodal, there exists at least one action $\tilde{a} \in \Pi_{G_\omega}(\cdot|s_t)$ such that $Q_\theta(s_t, \tilde{a}) \geq Q_\theta(s_t, \pi(\cdot|s_t))$.*

*Proof.* We first prove that there exists at least one action prior $\tilde{a} \in \Pi_{G_k(\cdot|s_t)}$ such that $Q_\theta(s_t, \tilde{a}) \geq Q_\theta(s_t, \pi_k(\cdot|s_t))$. We prove this by contradiction. Suppose for all $s_t, \tilde{a}$ that $Q_\theta(s_t, \tilde{a}) \leq Q_\theta(s_t, \pi_k(\cdot|s_t))$. Then, we have $Q_\theta(s_t, \pi_k(\cdot|s_t)) = Q_\theta(s_t, \pi^\star(\cdot|s_t))$. However, this contradicts the statement $Q_\theta(s_t, \pi_k(\cdot|s_t)) \leq Q_\theta(s_t, \pi_{k+1}(\cdot|s_t))$. Thus, this contradicts our claim about the existence of the action prior. We argue that within a finite number of training steps, there exists an action within the behavior policy support set from the generative network with a Q-value greater than that of the action from the current policy. $\square$

**Proposition B.4** (Noise-Regularized Q-Value Smoothness). *We assume that the action-value function $Q(s_t, a_t)$ is Lipschitz continuous concerning the action variable $a_t$ for any fixed state $s_t \in \mathcal{S}$. That is, there exists a constant $K_Q > 0$ such that for any $a_1, a_2 \in \mathcal{A}$,*

$$||Q(s_t, a_1) - Q(s_t, a_2)|| \leq K_Q \|a_1 - a_2\|. \tag{B.1}$$

*This condition ensures that small perturbations in the action space induce proportionally bounded variations in the Q-value, thereby promoting local smoothness of the critic function. We derive a precise upper bound on the approximation error*

$$\mathbb{E}_\epsilon |Q(s_t, a_t + \epsilon) - Q(s_t, a_t)| \approx \mathbb{E}_\epsilon |Q(s_t, a_t) + \nabla_{a_t} Q(s_t, a_t)^\top \epsilon + \frac{1}{2}\epsilon^\top \nabla_a^2 Q(s_t, \hat{a})\epsilon - Q(s_t, a_t)|$$

$$= \mathbb{E}_\epsilon |\nabla_{a_t} Q(s_t, a_t)^\top \epsilon + \frac{1}{2}\epsilon^\top \nabla_a^2 Q(s_t, \hat{a})\epsilon|$$

$$\leq \mathbb{E}_\epsilon |\nabla_{a_t} Q(s_t, a_t)^\top \epsilon| + \frac{1}{2}\mathbb{E}_\epsilon |\epsilon^\top \nabla_a^2 Q(s_t, \hat{a})\epsilon|, \tag{B.2}$$

*where $\hat{a} \in (a_t, a_t + \epsilon)$ is a point along the segment between $a_t$ and the noisy action $a_t + \epsilon$. For any state $s_t$ and action $a_t$, the Q-function $Q(s_t, a_t)$ is twice differentiable with respect to $a_t$, and there exists two constants $C_1 > 0, C_2 > 0$ such that*

$$\|\nabla_{a_t} Q(s_t, a_t)\| \leq C_1, \|\nabla_{a_t}^2 Q(s_t, a_t)\| \leq C_2. \tag{B.3}$$

We begin by analyzing the first term $\mathbb{E}_\epsilon|\nabla_{a_t}Q(s_t,a_t)^\top\epsilon|$. Since $\mathbb{E}[|X|] = 2\int_0^\infty x \cdot \frac{1}{\sqrt{2\pi}\tau}\exp\left(-\frac{x^2}{2\tau^2}\right)dx = \tau\sqrt{\frac{2}{\pi}}, X \sim \mathcal{N}(0,\tau^2)$ by applying Hölder's inequality (Folland, 1999), we obtain that

$$\mathbb{E}_\epsilon|\nabla_{a_t}Q(s_t,a_t)^\top\epsilon| \leq \mathbb{E}_\epsilon(||\nabla_{a_t}Q(s_t,a_t)||_\infty||\epsilon||_1) \leq C_1\mathbb{E}_\epsilon||\epsilon||_1 \leq C_1\sqrt{\frac{2}{\pi}}\tau. \qquad \text{(B.4)}$$

Next, we analyze the second term $\frac{1}{2}\mathbb{E}_\epsilon|\epsilon^\top\nabla_a^2 Q(s_t,\hat{a})\epsilon|$. According to the variance identity $\mathbb{E}[X^2] = \text{Var}(X) + (\mathbb{E}[X])^2$. In practice, we apply a symmetric clipping operation to the noise $\epsilon$, which ensures that $\text{Var}(clip(\epsilon)) \leq \tau^2$ and $\mathbb{E}(\epsilon) = 0$. It follows that

$$\mathbb{E}_\epsilon|\epsilon^\top\nabla_a^2 Q(s_t,\hat{a})\epsilon| \leq ||\nabla_a^2 Q(s_t,\hat{a})|| \cdot \mathbb{E}_\epsilon||\epsilon^2|| = C_2(\text{Var}(\epsilon) + 0) \leq C_2\tau^2. \qquad \text{(B.5)}$$

Consequently, the expected deviation is bounded by $\mathbb{E}_\epsilon|Q(s_t,a_t+\epsilon) - Q(s_t,a_t)| \leq C_1\sqrt{\frac{2}{\pi}}\tau + \frac{1}{2}C_2\tau^2$.

We observe that the upper bound of the approximation error increases as the policy variance grows. In the early stage of training, maintaining a low error bound is crucial for stabilizing Q-value estimation. In contrast, during the later stage, a larger noise variance is needed to preserve sufficient exploration and prevent the policy from converging to suboptimal solutions. Based on this observation, we propose an SDA Noise scheduling mechanism to enable adaptive noise modulation, thereby promoting more stable learning of Q-values.

## B.2 EFFICIENT POLICY LEARNING VIA POLICY DISTILLATION

To better understand the theoretical properties of our proposed method, we analyze its behavior in the tabular MDP setting. We provide convergence guarantees for policy evaluation, policy improvement, and policy iteration under our algorithm.

**Proposition B.5** (Policy Evaluation (Haarnoja et al., 2018))**.** *Let $\pi_k$ denote the policy at iteration $k$, and let $\pi_{k+1}$ be the greedy policy with respect to $Q^{\pi_k}$. Then, for all $(s_t,a_t) \in \mathcal{S} \times \mathcal{A}$, it holds that $Q^{\pi_{k+1}}(s_t,a_t) \geq Q^{\pi_k}(s_t,a_t)$.*

*Proof.* The noisy Bellman operator is defined as follows

$$\mathcal{T}_c^\pi Q(s_t,a_t) = r + \gamma\mathbb{E}_{s_{t+1}\sim\rho_\pi, a_{t+1}\sim\pi_\phi, \epsilon\sim\mathcal{N}(0,\tau^2)}\left[Q(s_{t+1},a_{t+1}+\epsilon) - \alpha\log\pi_\phi(a_{t+1}\mid s_{t+1})\right]. \tag{B.6}$$

Prove that the noise smoothing operator $\mathcal{T}_c^\pi$ is a contraction mapping. Let $Q_1, Q_2$ be any two functions, we have

$$\begin{aligned}|\mathcal{T}_c^\pi Q_1(s_t,a_t) - \mathcal{T}_c^\pi Q_2(s_t,a_t)| &= \gamma\left|\mathbb{E}_{s_{t+1},\epsilon}[Q_1(s_{t+1},\pi(\cdot|s_{t+1})+\epsilon) - Q_2(s_{t+1},\pi(\cdot|s_{t+1})+\epsilon)]\right| \\ &\leq \gamma\mathbb{E}_{s_{t+1},\epsilon}|Q_1(s_{t+1},\pi(\cdot|s_{t+1})+\epsilon) - Q_2(s_{t+1},\pi(\cdot|s_{t+1})+\epsilon)| \\ &\leq \gamma\|Q_1 - Q_2\|_\infty.\end{aligned} \tag{B.7}$$

Therefore, $\mathcal{T}_c^\pi$ is a $\gamma$-contractive mapping that satisfies the conditions of the banach fixed-point theorem and has a unique fixed point. We conclude that the Bellman operator $\mathcal{T}_c^\pi$ satisfies the $\gamma$-contraction property, which naturally leads to the conclusion that any initial Q function will converge to a unique fixed point by repeatedly applying $\mathcal{T}_c^\pi$. $\qquad\square$

**Proposition B.6** (Policy Improvement (Haarnoja et al., 2018))**.** *Let $\pi_k$ be the policy at iteration $k$, and $\pi_{k+1}$ be the updated policies, where $\pi_{k+1}$ is the greedy policy of the Q-value. Then for all $(s_t,a_t) \in \mathcal{S} \times \mathcal{A}, |\mathcal{A}| < \infty$, we have $Q^{\pi_{k+1}}(s_t,a_t) \geq Q^{\pi_k}(s_t,a_t)$.*

*Proof.* At iteration $k$, $\pi_k$ denotes the policy, and the corresponding value function is $Q^{\pi_k}$. We update the policy from $\pi_k$ to $\pi_{k+1}$, where $\pi_{k+1}$ is the greedy policy with respect to

$J_{\pi_k}(\pi)$, *i.e.*, $\pi_{k+1} = \arg\max_{\pi} \mathbb{E}_{a_t \sim \pi}[Q^{\pi_k}(s_t, a_t) + \alpha\mathcal{H}(\pi(a_t|s_t)) - \eta\mathrm{KL}(\pi_\phi(\cdot|s_t)||\tilde{a}_t)]$, where $\tilde{a} = \arg\max_{a}(Q_{\theta'}(s_t, a)), a \in \Pi_{G_\omega(\cdot|s_t)}$.

Since $\pi_{k+1} = \arg\max_{\pi} J_{\pi_k}(\pi)$, we have that $J_{\pi_k}(\pi_{k+1}) \geq J_{\pi_k}(\pi_k)$. Expressing $J_{\pi_k}(\pi_{k+1})$ and $J_{\pi_k}(\pi_k)$ by their definition, we have $\mathbb{E}_{a \sim \pi_{k+1}}[Q^{\pi_k}(s_t, a_t) + \alpha\mathcal{H}(\pi_{k+1}(a_t|s_t)) - \mu\mathrm{KL}(\pi_\phi(\cdot|s_t)||\tilde{a}_t)] \geq \mathbb{E}_{a \sim \pi_k}[Q^{\pi_k}(s_t, a_t) + \alpha\mathcal{H}(\pi_k(a|s)) - \mu\mathrm{KL}(\pi_\phi(\cdot|s_t)||\tilde{a}_t)]$.

In a similar way to the proof of the soft policy improvement (Sutton et al., 1998), we come to the following inequality:

$$
\begin{aligned}
Q^{\pi_k}(s_t, a_t) =& r(s_t, a_t) + \gamma\mathbb{E}_{s_{t+1} \sim \mathcal{B}, a_{t+1} \sim \pi_k}[Q^{\pi_k}(s_{t+1}, a_{t+1} + \epsilon) + \alpha\mathcal{H}(\pi_k(a_{t+1}|s_{t+1}))] \\
\leq& r(s_t, a_t) + \gamma\mathbb{E}_{s_{t+1} \sim \mathcal{B}, a_{t+1} \sim \pi_{k+1}}[Q^{\pi_k}(s_{t+1}, a_{t+1} + \epsilon) + \alpha\mathcal{H}(\pi_{k+1}(a_{t+1}, s_{t+1}))] \\
&\vdots \\
\leq& Q^{\pi_{k+1}}(s_t, a_t).
\end{aligned}
$$
(B.8)

Here, the inequality is obtained by repeatedly expanding $Q^{\pi_k}$ on the right-hand side of the equation through $Q^{\pi_k}(s_t, a_t) = r(s_t, a_t) + \gamma\mathbb{E}_{s_{t+1}}[\mathbb{E}_{a_{t+1} \sim \pi_k}[Q^{\pi_k}(s_{t+1}, a_{t+1}) + \alpha\mathcal{H}(\pi_k(\cdot|s_{t+1}))]]$ and applying the inequality $\mathbb{E}_{a \sim \pi_{k+1}}[Q^{\pi_k}(s_t, a_t) + \mathcal{H}(\pi_{k+1}(\cdot|s_t)] \geq \mathbb{E}_{a \sim \pi_k}[Q^{\pi_k}(s_t, a_t) + \mathcal{H}(\pi_k(\cdot|s_t)]$. Finally, we arrive at convergence to $Q^{\pi_{k+1}}(s_t, a_t)$ and finish the proof. $\square$

**Proposition B.7** (Policy Iteration (Haarnoja et al., 2018)). *By alternating policy evaluation and policy improvement steps, the resulting sequence of policies $\{\pi\}$ converges to an optimal policy $\pi^\star$ such that $Q^{\pi^\star}(s_t, a_t) \geq Q^{\pi_{k+1}}(s_t, a_t), \forall\pi_{k+1} \in \Pi, \forall(s_t, a_t) \in \mathcal{S} \times \mathcal{A}$.*

*Proof.* Let $\Pi$ be the space of policy distributions and let $\pi_k$ be the policies at iteration $k$. By the policy improvement property in Proposition B.6, the sequence $Q^{\pi_k}$ is monotonically increasing. Also, for any state-action pair $(s_t, a_t) \in \mathcal{S} \times \mathcal{A}$, each $Q^{\pi_k}$ is bounded due to the discount factor $\gamma$. Thus, the sequence of $\pi_k$ converges to some $\pi^\star$ that is a local optimum. We assume a finite MDP, as typically assumed for the convergence proof in usual policy iteration (Sutton et al., 1998). At convergence, we get $J_{\pi^\star}(\pi^\star)[s_t] \geq J_{\pi^\star}(\pi_{k+1})[s_t], \forall\pi_{k+1} \in \Pi$. Using the same iterative augment as in the proof of Proposition B.6, we get $Q^{\pi^\star}(s_t, a_t) \geq Q^{\pi_{k+1}}(s_t, a_t)$ for all $(s_t, a_t) \in \mathcal{S} \times \mathcal{A}$. Hence, $\pi^\star$ are optimal in $\Pi$. $\square$

## C    EXPERIMENTAL SETUP AND IMPLEMENTATION DETAILS

### C.1    THREE CONTINUOUS CONTROL ENVIRONMENTS

We evaluated the effectiveness of the B2PD algorithm in seven MuJoCo (Todorov et al., 2012) environments, four PyBullet (Coumans & Bai, 2016) environments, and four DMControl (Tassa et al., 2018) environments, including both state-based and pixel-based settings, as shown in Figure C.1. We provided a detailed overview of these environments.

**MuJoCo.** MuJoCo is a fast and powerful physics engine that has become a standard platform for simulating complex robotic dynamics and control tasks. It is extensively utilized in the robotics and RL communities for training agents to perform tasks such as locomotion and manipulation, and it offers a diverse set of benchmarks for evaluating RL algorithms. In our experiments, we adopt seven challenging tasks from the MuJoCo environments: HalfCheetah, Ant, Hopper, Walker2d, Humanoid, InvertedDoublePendulum, and InvertedPendulum.

**PyBullet.** PyBullet is an open-source physics engine that supports general-purpose physical simulation as well as high-fidelity modeling for robotics tasks. It provides an efficient control interface for fast simulation of rigid-body dynamics, collision detection, and various control problems. In this work, we evaluated B2PD on four PyBullet benchmark tasks: HalfCheetahBulletEnv, AntBulletEnv, HopperBulletEnv, and Walker2DBulletEnv.

**DMControl.** The DeepMind Control (DMControl) suite is a set of continuous control environments mainly designed for robotic manipulation tasks, but it also covers a wide range of industrial control

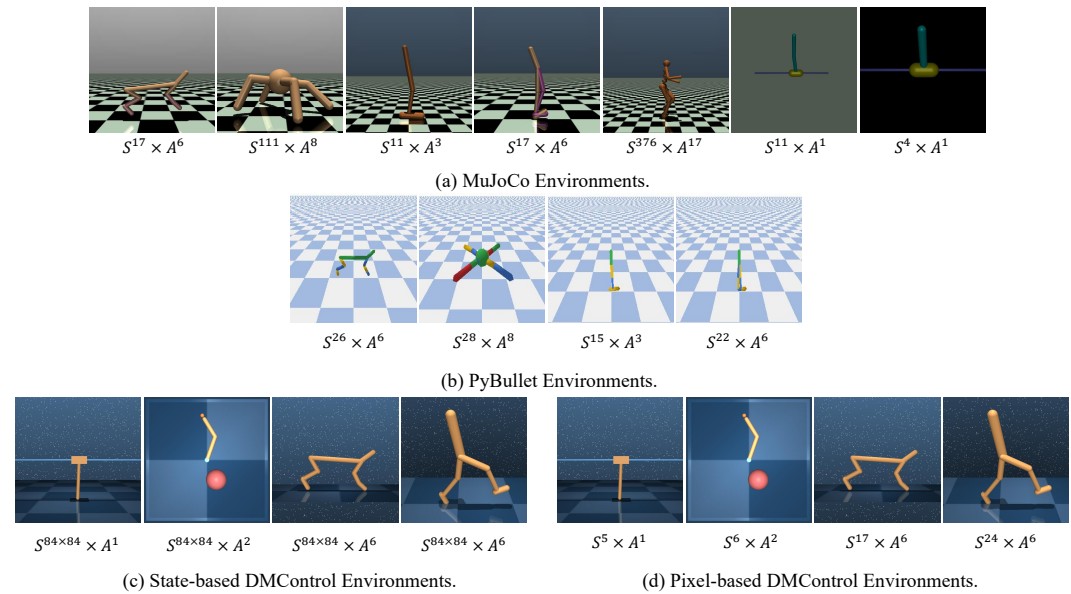

$S^{17} \times A^6$  $S^{111} \times A^8$  $S^{11} \times A^3$  $S^{17} \times A^6$  $S^{376} \times A^{17}$  $S^{11} \times A^1$  $S^4 \times A^1$

(a) MuJoCo Environments.

$S^{26} \times A^6$  $S^{28} \times A^8$  $S^{15} \times A^3$  $S^{22} \times A^6$

(b) PyBullet Environments.

$S^{84 \times 84} \times A^1$  $S^{84 \times 84} \times A^2$  $S^{84 \times 84} \times A^6$  $S^{84 \times 84} \times A^6$   $S^5 \times A^1$  $S^6 \times A^2$  $S^{17} \times A^6$  $S^{24} \times A^6$

(c) State-based DMControl Environments.          (d) Pixel-based DMControl Environments.

Figure C.1: Images of the MuJoCo, PyBullet, and DMControl environments used in our experiments, including 7 MuJoCo continuous control tasks, 4 PyBullet continuous control tasks, 4 state-based DMControl continuous control tasks and 4 pixel-based DMControl continuous control tasks.

challenges. In our experiments, we evaluated the proposed algorithm on four continuous control tasks from the DMControl suite: cartpole-swingup, reacher-easy, cheetah-run, and walker-walk. Each task was assessed under both state-based and pixel-based inputs, with identical action spaces; the only difference lies in the form of the observations.

## C.2 Detailed Description of Baselines and B2PD Algorithms

**Software.** We use the following software versions:

- Python 3.9
- Pytorch 2.5.0
- Gym 0.26.2
- MuJoCo 3.3.1
- MuJoCo-py 2.1.2.14
- PyBullet 3.2.7
- dm_control 1.0.28

**Detailed Description of the NNPG Algorithm.** We implement the nearest-neighbor exploration algorithm proposed in NNAC (Shen & Yang, 2021) based on the SAC algorithm. NNPG employs the nearest neighbor action from the online replay buffer as an anchor for the current policy decision and adopts an online policy distillation paradigm to guide the actor update. The overall actor loss can be formulated as

$$J_\pi(\phi) = \mathbb{E}_{s_t \sim \mathcal{B}} \left[ Q_\theta(s_t, \pi_\phi(\cdot|s_t)) + \alpha \mathcal{H}\left(\pi_\phi(\cdot|s_t)\right) - \eta \mathrm{KL}(\pi_\phi(\cdot|s_t)||\tilde{a}) \right]. \quad (C.1)$$

The nearest neighbor action $\tilde{a}$ is queried from the replay buffer $\mathcal{B}_a$, which can be formulated as

$$\tilde{a} = \arg\min_a(||\pi_\phi(\cdot|s_t) - \mathcal{B}_a||_\infty) \quad (C.2)$$

where $|| \cdot ||_\infty$ denotes the Chebyshev distance (Deza et al., 2009).

---

**Algorithm 1** Bidirectional Behavior Prior Distillation

---

Initialize behavior policy network $G_\omega$, critic networks $Q_\theta$ and actor-network
$\pi_\phi$ with random parameters $\omega$, $\theta$, $\phi$
Initialize target networks $\theta' \leftarrow \theta$, $\phi' \leftarrow \phi$
Initialize empty replay buffer $\mathcal{B}$
**for** $t = 0$ to $T$ **do**
    Calculate action $a \sim \pi_\phi(\cdot|s_t)$
    Get reward $r_t$ and new state $s_{t+1}$
    Store transition tuple $(s_t, a_t, r_t, s_{t+1})$ in $\mathcal{B}$
    Sample mini-batch of transitions $(s_t, a_t, r_t, s_{t+1}) \sim \mathcal{B}$
    Train critic:
    Update critic by minimizing Eq. 6
    Train behavior prior network:
    Update CVAE by minimizing Eq. 11
    Train actor:
    Get the expert policy anchor $\tilde{a}$ by Eq. 12
    Update actor by maximizing Eq. 14
    Update weights:
    $\theta' \leftarrow \lambda\theta + (1 - \lambda)\theta'$
    $\phi' \leftarrow \lambda\phi + (1 - \lambda)\phi'$
**end for**

---

**Detailed Description of the B2PD Algorithm.** We constructed the B2PD algorithm using SAC, as detailed in Algorithm 1. First, we propose CVAE to learn policy priors from the online replay buffer and dynamically generate a high-value support set policy prior $\Pi_{G_\omega(s_t)}$. Subsequently, we introduce the behavior policy prior distillation mechanism, which selects high-reward actions as anchors to guide the policy update process, thereby significantly improving the sample efficiency of online RL. To further improve stability, we propose an SDA Noise mechanism to strengthen Q-value estimation, providing action-value priors for the CVAE. Finally, the weights of the Critic and Actor networks are updated utilizing an exponential moving average (EMA) to ensure stable network updates.

Our proposed B2PD algorithm is implemented based on SAC, and to ensure fair comparison, we adopt the hyperparameters suggested in the original paper. B2PD involves three key hyperparameters: action value prior coefficient in CVAE $\xi$, the number of policy prior $H$, and behavior policy prior coefficient $\eta$. For all tasks, we set $\xi$=0.1, $H$=10, and $\eta$=1.0. A comprehensive list of hyperparameters for B2PD, B2PD (TD3), and five state-based baselines is provided in Table C.1. Hyperparameters used for B2PD-RAD and vanilla RAD are reported in Table C.2, while those for B2PD-DrQ-v2 and vanilla DrQ-v2 are listed in Table C.3.

**CVAE network structural design.** Our behavior cloning model architecture is designed based on the CVAE framework proposed in BCQ (Fujimoto et al., 2019) and MCQ (Lyu et al., 2022). The complete configuration of model parameters is detailed in Table C.4, which systematically presents the dimensional settings of the encoder, decoder, and the latent space.

# D   MORE EXPERIMENTS ON TOY ENVIRONMENT

To further analyze the policy improvement process of B2PD, we also visualize the predicted actions at timesteps 30K, 50K, 100K, 150K, and 200K, as shown in Figure D.1. We observe that the predicted action directions for proximal states are consistent, indicating strong stability in the policy updates of B2PD. B2PD learns a coherent action field, with directions consistently pointing toward high-reward regions. A structured pattern emerges early in training and becomes increasingly pronounced over time. Together, these observations demonstrate that B2PD enables the agent to learn efficiently and stably by leveraging behavior priors. It achieves strong policy convergence and action consistency, outperforming standard maximum entropy policies in environments with complex reward structures.

| Hyper-parameter | DDPG | TD3 | SAC | ALH | NNPG | B2PD (TD3) | B2PD |
|---|---|---|---|---|---|---|---|
| Optimizer | Adam | Adam | Adam | Adam | Adam | Adam | Adam |
| Critic learning rate | $3 \cdot 10^{-4}$ | $3 \cdot 10^{-4}$ | $3 \cdot 10^{-4}$ | $3 \cdot 10^{-4}$ | $3 \cdot 10^{-4}$ | $3 \cdot 10^{-4}$ | $3 \cdot 10^{-4}$ |
| Actor learning rate | $3 \cdot 10^{-4}$ | $3 \cdot 10^{-4}$ | $3 \cdot 10^{-4}$ | $3 \cdot 10^{-4}$ | $3 \cdot 10^{-4}$ | $3 \cdot 10^{-4}$ | $3 \cdot 10^{-4}$ |
| Target update rate | $5 \cdot 10^{-3}$ | $5 \cdot 10^{-3}$ | $5 \cdot 10^{-3}$ | $5 \cdot 10^{-3}$ | $5 \cdot 10^{-3}$ | $5 \cdot 10^{-3}$ | $5 \cdot 10^{-3}$ |
| Batch size | 256 | 256 | 256 | 256 | 256 | 256 | 256 |
| Discount factor | 0.99 | 0.99 | 0.99 | 0.99 | 0.99 | 0.99 | 0.99 |
| Number of critics | 1 | 2 | 2 | 2 | 2 | 1 | 2 |
| Policy update frequency $d$ | 1 | 2 | 1 | 2 | 1 | 1 | 1 |
| Exploration noise | - | $\mathcal{N}(0,0.1)$ | - | $\mathcal{N}(0,0.1)$ | - | $\mathcal{N}(0,0.1)$ | - |
| Policy noise $\epsilon$ | $\mathcal{N}(0,0.2)$ | $\mathcal{N}(0,0.2)$ | - | $\mathcal{N}(0,0.2)$ | - | $\mathcal{N}(0,0.2)$ | Eq. 8 |
| Noise clip range | [-0.5,0.5] | [-0.5,0.5] | - | [-0.5,0.5] | - | [-0.5,0.5] | [-0.5,0.5] |
| Action value prior coefficient $\xi$ | - | - | - | - | - | 0.1 | 0.1 |
| Number of policy prior $H$ | - | - | - | - | - | 10 | 10 |
| Behavior policy prior coefficient $\eta$ | - | - | - | - | 1.0 | 1.0 | 1.0 |
| Critic | (state dim.+ action dim.)-256-256-1 | | | | | | |
| Actor | state dim.-256-256-action dim. | | | | | | |

Table C.1: A complete comparison of hyper-parameter choices between B2PD, B2PD (TD3) and five baselines, including DDPG, TD3, SAC, ALH, and NNPG.

| | Hyperparameter | Value |
|---|---|---|
| RAD | Augmentation | Crop - walker, walk; Translate - otherwise |
| | Observation rendering | $(100, 100)$ |
| | Observation down/upsampling | $(84, 84)$ (crop); $(108, 108)$ (translate) |
| | Replay buffer size | 100000 |
| | Initial steps | 1000 |
| | Stacked frames | 3 |
| | Action repeat | walker, walk |
| | | 8 cartpole, swingup |
| | | 4 otherwise |
| | Hidden units (MLP) | 1024 |
| | Evaluation episodes | 10 |
| | Optimizer | Adam |
| | Learning rate | $1e - 4$ |
| | Batch Size | 512 |
| | $Q$ function EMA | 0.01 |
| | Critic target update freq | 2 |
| | Convolutional layers | 4 |
| | Number of filters | 32 |
| | Non-linearity | ReLU |
| | Encoder EMA | 0.05 |
| | Latent dimension | 50 |
| | Discount $\gamma$ | .99 |
| | Initial temperature | 0.1 |
| B2PD | Policy noise $\epsilon$ | Eq. 8 |
| | Action value prior coefficient $\xi$ | 0.1 |
| | Number of policy prior $H$ | 10 |
| | Behavior policy prior coefficient $\eta$ | 1.0 |

Table C.2: Hyperparameters employed for RAD and B2PD-RAD in the DMControl experiments.

# E    MORE EXPERIMENTS ON MUJOCO ENVIRONMENT

## E.1    COMPARISON OF DISTILLATION LOSS FOR B2PD

To constrain the deviation between the current actor's decisions and the expert prior, we draw inspiration from knowledge distillation (Sun et al., 2024) by introducing commonly used KL diver-

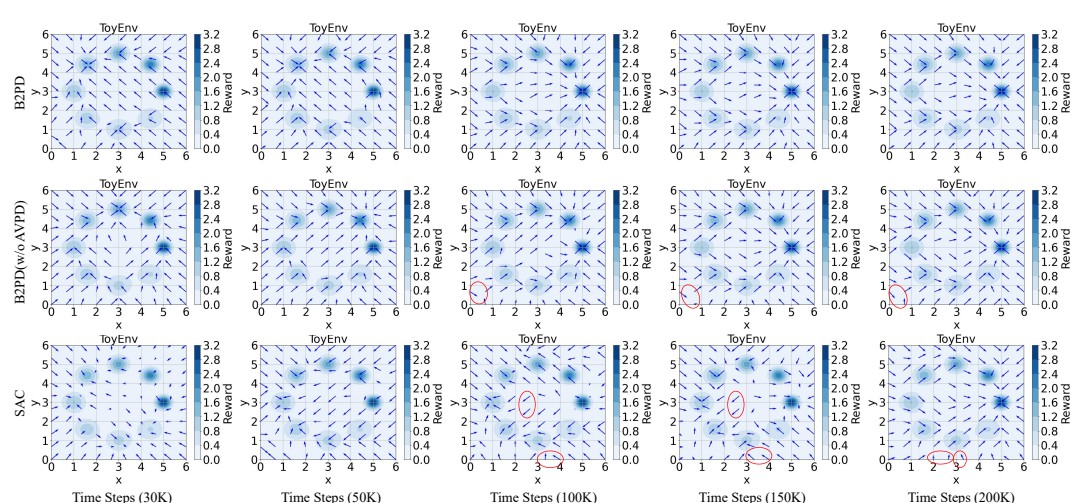

Figure D.1: Action prediction results visualized in a state space composed of eight Gaussian mixture distributions. Blue arrows indicate the predicted action directions, with their lengths representing the action magnitudes. Several low-quality decisions observed between 100K and 200K environment steps are highlighted with red circles.

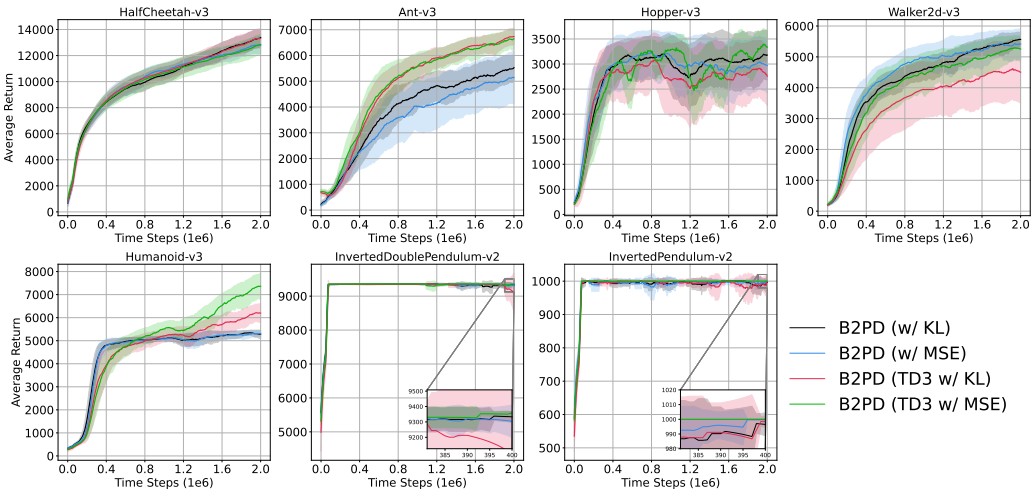

Figure E.1: Learning curves on seven MuJoCo continuous control tasks. The shaded area captures a 95% confidence interval around the average performance over 10 trials. Curves are smoothed uniformly for visual clarity.

| | Hyper-parameter | Setting |
|---|---|---|
| DrQ-v2 | Replay buffer capacity | $10^6$ |
| | Action repeat | 2 |
| | Seed frames | 4000 |
| | Exploration steps | 2000 |
| | $n$-step returns | 3 |
| | Mini-batch size | 256 |
| | Discount $\gamma$ | 0.99 |
| | Optimizer | Adam |
| | Learning rate | $10^{-4}$ |
| | Agent update frequency | 2 |
| | Critic Q-function soft-update rate | 0.01 |
| | Features dim. | 50 |
| | Hidden dim. | 1024 |
| | Exploration stddev. clip | 0.3 |
| | Exploration stddev. schedule | $\mathrm{linear}(1.0, 0.1, 100000)$ |
| B2PD | Policy noise $\epsilon$ | Eq. 8 |
| | Action value prior coefficient $\xi$ | 0.1 |
| | Number of policy prior $H$ | 10 |
| | Behavior policy prior coefficient $\eta$ | 1.0 |

Table C.3: Hyperparameters employed for DrQ-v2 and B2PD-DrQ-v2 in the DMControl experiments.

| **CVAE Hyperparameters** | |
|---|---|
| Optimizer | Adam |
| CVAE learning rate | $3 \times 10^{-4}$ |
| $z$ dimension | $2\cdot$ action dimension |
| Hidden activation function | ReLU Layer |
| Encoder hidden dimension | 750 |
| Decoder hidden dimension | 750 |
| Hidden layers | 2 |

Table C.4: Hyperparameter setup for CVAE.

gence (Hinton et al., 2015) and L2 distance metrics. The experimental results are shown in Figure E.1.

It can be observed that using KL divergence as the optimization objective leads to a significant performance gain in the B2PD framework, while MSE loss achieves superior results under the B2PD (TD3) setting. We attribute this difference to the distinct nature of the policy structures: KL divergence loss provides greater stability for stochastic policies parameterized by Gaussian distributions, whereas MSE loss offers more direct and explicit supervision for deterministic policy updates. This finding highlights the importance of selecting distillation objectives that are aligned with the underlying policy structure, thereby enabling more efficient knowledge transfer and policy convergence.

### E.2 COMPARISON WITH ADDITIONAL NOISE SCHEDULING METHODS

Since noise injection has been shown to stabilize Q-value estimation, we systematically evaluate the effectiveness of different noise-scheduling strategies in both the B2PD and SAC frameworks. Specifically, we compare B2PD with SDA noise, B2PD with SDA noise but without policy-entropy regularization (w/ SDA Noise, w/o Policy Entropy), and several alternative noise-scheduling approaches, including standard Gaussian noise (Fujimoto et al., 2018), linearly decayed Gaussian noise (Hollenstein et al., 2022), and our proposed SDA mechanism. To further isolate and understand the role of noise-scheduling mechanisms within SAC, we additionally apply the same set

of noise-scheduling strategies to the vanilla SAC algorithm. All configurations are trained for 2M environment steps using 10 random seeds. The detailed results are summarized in Figure E.2.

As shown in Figure E.2, B2PD (w/ Linear Decay Gaussian Noise) performs worse than the standard B2PD configuration. This degradation suggests that the progressive reduction of noise magnitude during training, imposed by the linear decay schedule, may weaken the smoothing effect that is critical for stable Q-value estimation. Furthermore, both B2PD and SAC (w/ SDA Noise) respectively outperform B2PD (w/ Gaussian Noise) and SAC (w/ Gaussian Noise), highlighting the effectiveness of variance-aware noise modulation. Notably, B2PD (w/ SDA Noise, w/o Policy Entropy)

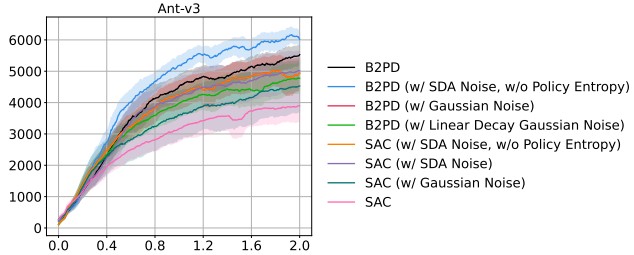

Figure E.2: Comparison of the impact of different policy noises on the performance of the B2PD algorithm. The shaded area represents the 95% confidence interval over the average performance of 10 trials. Curves are uniformly smoothed for visual clarity.

achieves higher returns than the default B2PD variant. In contrast, SAC (w/ SDA Noise, w/o Policy Entropy) shows no clear improvement over SAC (w/ SDA Noise). This discrepancy indicates that, given SAC's inherently high-entropy exploration mechanism, combining policy-entropy regularization with SDA noise provides more accurate value guidance. By comparison, B2PD naturally operates in a low-entropy regime, where the SDA noise mechanism alone is sufficient to ensure stable and accurate value estimation.

### E.3    INTEGRATING B2PD WITH TD7

To further assess the effectiveness of our proposed method, we combine B2PD with the recently introduced TD7 algorithm (Fujimoto et al., 2023) and conduct experiments across standard MuJoCo continuous-control benchmarks. All hyperparameters follow the unified configuration in Table C.1, and each setting is evaluated over 10 random seeds for statistical robustness. The learning curves in Figure E.3 show that incorporating B2PD consistently improves the performance of TD7 across all four tasks.

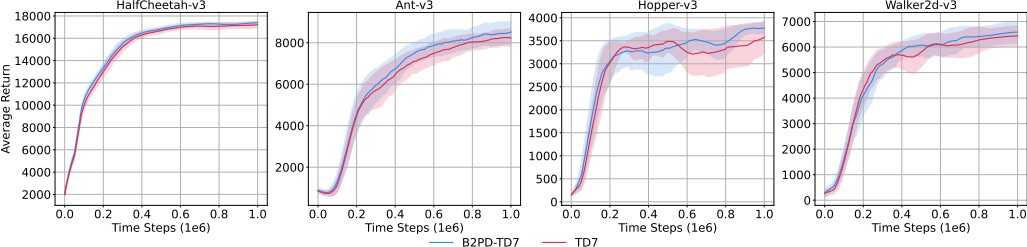

Figure E.3: Learning curves on four MuJoCo continuous control tasks using TD7 as the baseline. The shaded area captures a 95% confidence interval around the average performance over 10 trials. Curves are smoothed uniformly for visual clarity.

### E.4    DOES THE BEHAVIORAL POLICY NETWORK GENERATE HIGH-VALUE ACTION PRIORS?

To evaluate whether the behavioral policy network can generate high-value policies, we evaluate the immediate rewards of actions sampled from both the policy prior network $G_\omega$ and the actor network $\pi_\phi$, each trained for 300K environment steps under the B2PD algorithm. The evaluations are performed using the physical environment model $\mathcal{P}(\cdot|s_t, a_t)$ provided by Gym (Brockman et al., 2016), and their reward distributions are visualized in Figure E.4.

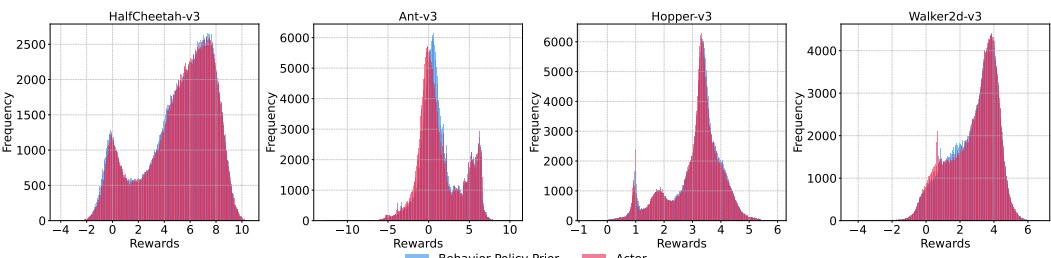

Figure E.4: Reward distributions for four MuJoCo continuous control tasks, visualized using 300-bin histograms.

As the Figure E.4 shows, the action value distributions from the policy prior network and the actor network exhibit a similar trend. Notably, the value distribution for actions from the policy prior network tends to be more concentrated in the high-value region. This indicates that using a CVAE as the behavioral policy network enables the generation of high-value episodes, thereby providing the agent with a more directive policy prior. This enables the agent to explore high-value actions more efficiently during subsequent updates and reduces ineffective exploration. This is consistent with the efficacy of policy distillation observed in the toy experiments in Appendix D.

# F   MORE EXPERIMENTS ON PYBULLET ENVIRONMENT

## F.1   MAIN RESULTS ON PYBULLET ENVIRONMENT

We evaluate B2PD by combining it with TD3 and SAC, and compare it against IR-SAC, where intrinsic motivation is implemented following (Chentanez et al., 2004). We conducted experiments on the PyBullet benchmark suite, training each algorithm for 2 million timesteps using 10 random seeds. The learning curves are summarized in Figure F.1, and the final performance is reported in Table F.1.

Across all four PyBullet tasks, B2PD and B2PD (TD3), built upon SAC and TD3 respectively, consistently outperform their vanilla counterparts. These substantial performance gains demonstrate that B2PD can effectively enhance the practical efficiency of online RL algorithms. In contrast, IR-SAC exhibits only moderate improvements on HalfCheetahBulletEnv and Walker2DBulletEnv, while offering limited benefits over vanilla SAC on the remaining tasks. We hypothesize that intrinsic rewards may induce excessive exploration, leading to non-stationary training dynamics that hinder stable policy improvement.

| Tasks | DDPG | TD3 | SAC | IR-SAC | B2PD (TD3) | B2PD |
|---|---|---|---|---|---|---|
| HalfCheetahBulletEnv-v0 | 693±235 | 1820±482 | 2015±246 | 2231±188 | **2863±247** | 2524±195 |
| AntBulletEnv-v0 | 170±131 | 2423±140 | 2216±230 | 2176±249 | **3338±190** | 3091±131 |
| HopperBulletEnv-v0 | 1586±365 | 2082±157 | 2075±374 | 2106±450 | **2326±153** | 2123±214 |
| Walker2DBulletEnv-v0 | 583±153 | 1819±247 | 1480±228 | 1736±174 | 2390±166 | **2411±149** |

Table F.1: Average return of the last 10 evaluation scores on Pybullet environments over 10 random seeds, where ± captures a 95% confidence interval. The maximum values of each row are bolded.

## F.2   HYPERPARAMETER STUDY

B2PD introduces three hyperparameters to control the distillation strength: the action-value prior distillation coefficient $\xi$, the number of sampled behavioral priors $H$, and the behavior prior distillation coefficient $\eta$. To demonstrate the parameter sensitivity of B2PD, we trained for timesteps on four PyBullet tasks over 10 different random seeds.

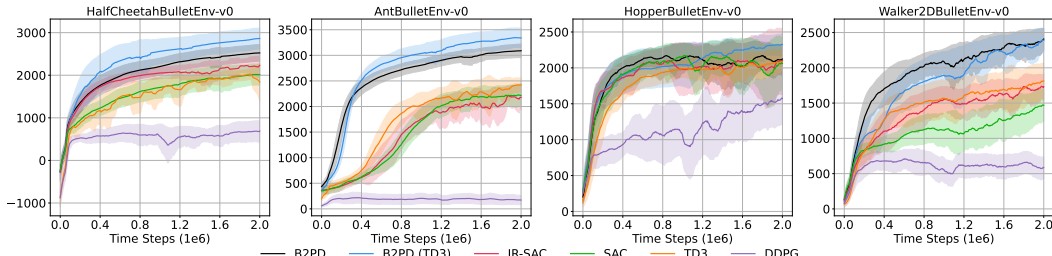

Figure F.1: Learning curves on four PyBullet continuous control tasks. The shaded area captures a 95% confidence interval around the average performance over 10 trials. Curves are smoothed uniformly for visual clarity.

**Action-value prior distillation coefficient** $\xi$. The action-value prior distillation weight $\xi$ is a critical hyperparameter in B2PD, as it directly controls the balance between reconstruction and high-value generation in the CVAE's training objective. When $\xi=0$, the algorithm degenerates to B2PD (w/o AVPD). Intuitively, one should not use a large $\xi$ to ensure the stability of CVAE training. We observe from Figure F.2 that while a larger $\xi$ can improve action quality during training, it may also lead to potential training instability, particularly on the HopperBulletEnv and Walker2DBulletEnv tasks.

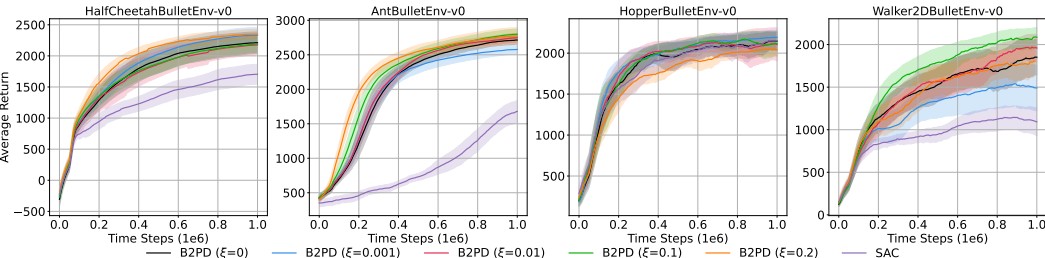

Figure F.2: Sensitivity analysis with respect to parameter $\xi$. The shaded area captures a 95% confidence interval around the average performance over 10 trials. Curves are smoothed uniformly for visual clarity.

**Number of sampled behavioral priors** $H$. When the behavioral prior policy $G$ is known, $H$ should be set to a sufficiently large value to optimally sample actions with the highest Q-value. However, in our practical setting, we use a CVAE to approximate the support set of the behavior policy prior and sample high-value actions from it. Here, $H$ takes on the role of balancing optimistic optimization and constrained exploration. To explore the influence of $H$, we conducted experiments on four tasks with a fixed hyperparameter $\xi=0.1$. As observed from Figure F.3, the B2PD algorithm is insensitive to changes in $H$ for the HalfCheetahBulletEnv and AntBulletEnv tasks. Based on this empirical evidence, we set the default value of $H$ to 10 across all experiments.

**Behavior prior distillation coefficient** $\eta$. The behavioral prior distillation coefficient $\eta$ controls the balance between prior policy distillation and the maximum entropy optimization objective. When $\eta=0$, the B2PD algorithm degenerates to the vanilla SAC algorithm combined with SDA noise, which improves performance on AntBulletEnv, HopperBulletEnv, and Walker2DBulletEnv tasks. As observed from the Walker2DBulletEnv task in Figure F.4, a higher performance is achieved with $\eta=1.0$. For other tasks, however, performance gains were observed with different values of $\eta$, suggesting that task-specific parameter tuning is required to achieve better performance.

**Can the B2PD algorithm perform policy distillation under noisy priors?** In online RL, if the behavioral priors concentrate in low-value regions, they exacerbate the sample-efficiency problem and undermine the effectiveness of policy distillation. To assess the robustness of B2PD under such conditions, we uniformly inject state noise during the inference phase of the prior, defined as $s_t^\star = s_t + \mathcal{N}(0, \epsilon_s^2)$, where $\epsilon_s \in \{0.1, 0.3, 1.0\}$. We construct two versions of behavioral prior

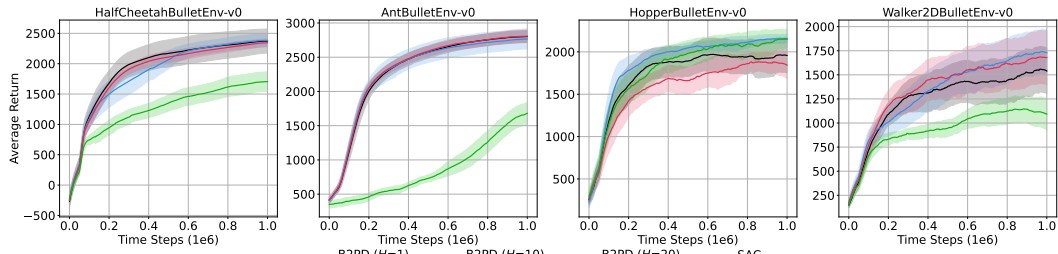

Figure F.3: Sensitivity analysis with respect to parameter $H$. The shaded area captures a 95% confidence interval around the average performance over 10 trials. Curves are smoothed uniformly for visual clarity.

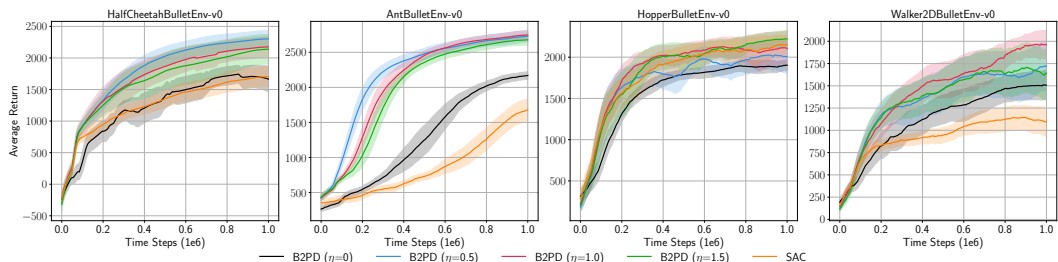

Figure F.4: Sensitivity analysis with respect to parameter $\eta$. The shaded area captures a 95% confidence interval around the average performance over 10 trials. Curves are smoothed uniformly for visual clarity.

distributions. The first is inferred from noisy states, where Gaussian noise with scale $\epsilon_s$ is injected into the state inputs. The second is inferred from clean states without corruption. Given these priors, we use the Q-network to select the expert policy anchor, formalized as

$$\tilde{a} = \arg \max_{a_h} \left( Q_{\theta'}\left(s_t, a_h\right)\right), a_h \in \Pi_{G_\omega(\cdot|\{s_t, s_t^\star\})}. \tag{F.1}$$

To further assess whether B2PD can filter out high-quality priors even when the action support itself becomes unreliable, we inject action noise $\epsilon_a$ into the action support set. This noise is applied to each action proposal before anchor selection. The noisy-anchor filtering process is then expressed as

$$\tilde{a} = \arg \max_{a \in \{a_h, a_h^\star\}} Q_{\theta'}(s_t, a), a_h \sim \Pi_{G_\omega}(\cdot|s_t), a_h^\star = a_h + \epsilon_a. \tag{F.2}$$

We conduct experiments on the HalfCheetahBulletEnv and AntBulletEnv tasks for 1M timesteps, and report results averaged over 10 random seeds. The corresponding performance curves are shown in Figure F.5.

Figure F.5 demonstrate that B2PD consistently benefits from Gaussian state noise injected at three different intensity levels, and action noise further contributes to performance improvements. This improvement arises from two complementary effects: first, the proposed SDA Noise mechanism stabilizes Q-value learning, which helps to extract high-quality policy anchors; second, the injected noise increases the diversity of the inferred action priors, providing the actor with a richer set of high-value actions. Notably, these noise-conditioned priors differ fundamentally from actions sampled from clean states: the former correspond to neighboring strategies of the current policy, capturing its potential future trends, whereas the latter may include more out-of-distribution actions. As a result, priors inferred from noisy states exhibit stronger robustness, allowing B2PD to achieve higher sample efficiency and faster policy convergence in purely online RL settings. These results further underscore the promise of B2PD for complex, real-world scenarios.

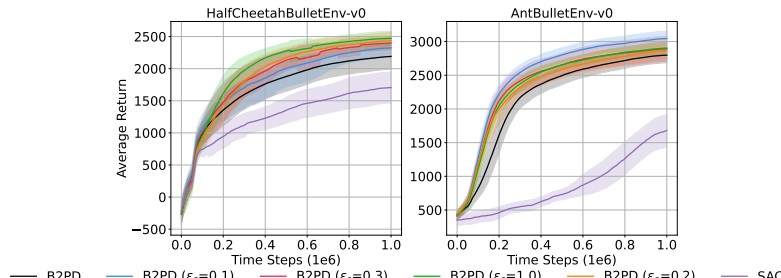

Figure F.5: Average return over 10 seeds with different levels of conditional state-noise and action-noise applied to the behavioral prior. Curves are smoothed uniformly for visual clarity.

## F.3 RUNTIME

We conducted experiments in a Linux environment equipped with a 56-core Xeon(R) 6133 CPU and 1 Nvidia RTX 3090 GPU to evaluate the computational time of our proposed B2PD method against DDPG, TD3, SAC, ALH, and NNPG. As illustrated in Table F.2, our B2PD method significantly increased the training time compared to vanilla algorithms, with both B2PD and B2PD (TD3) showing considerable time consumption. This additional computational overhead is mainly due to the training and inference procedures of the CVAE.

Table F.2: Wall-clock training time of each algorithm on the HalfCheetah-v3 task, measured over 2M environment steps.

| Methods | DDPG | TD3 | SAC | ALH | NNPG | B2PD (TD3) | B2PD |
|---------|------|-----|-----|-----|------|------------|------|
| runtime | 8.1h | 5.4h | 6.3h | 6.8h | 11.5h | 10.0h | 11.4h |

## G THE USE OF LARGE LANGUAGE MODELS (LLMS)

We used a large language model (GPT-5) solely for language polishing during manuscript preparation. The LLM was not involved in research idea generation, experimental design, data analysis, or drawing scientific conclusions. All content and claims in the paper are the responsibility of the authors.

## H POTENTIAL NEGATIVE SOCIETAL IMPACTS

In this work, we propose a bidirectional behavioral prior distillation algorithm to improve the performance of online RL. It is worth noting that any potential negative social impacts are associated with traditional reinforcement learning algorithms. We emphasize the importance of adhering to fair and safe practices for applications in industry control, gaming AI, and other areas.

