# OpenReview forum: "Improving Online Reinforcement Learning via Behavior Prior Distillation"
_ICLR.cc/2026/Conference — Submitted to ICLR 2026_

### Official Review · Reviewer_eZE4 · 2025-10-26

**Soundness:** 1
**Presentation:** 2
**Contribution:** 2
**Rating:** 2
**Confidence:** 3

**Summary:**

This work proposes an off-policy actor critic algorithm for online reinforcement learning. Departing from the standard approach of distilling a prior from offline data, the method relies on a behavior prior distilled from the data collected online, and guided by the critic. In practice, the algorithm seems to combine different techniques and ideas:
- entropy regularization as in SAC
- a multi-modal behavior prior (CVAE)
- a random-shooting procedure for computing action targets while distilling the prior
- noise injection over actions as in TD3

The combination of these techniques produces the proposed algorithm, B2PD, which is evaluated extensively in state-based and visual online RL (across standard Mujoco and Pybullet continuous control environments). The submission is concluded by an ablation visualizing entropy curves on a single task, and by a detailed Appendix.

**Strengths:**

- The goal of designing a sample efficient online RL algorithm remains relevant.
- The experimental evaluation is sufficiently broad and includes several relevant baselines.
- Empirically, the method seems to perform rather well, at a modest computational overhead.

**Weaknesses:**

- The core idea of the method is not justified formally, to the best of my understanding. Despite a few propositions, it's unclear why one should train both a policy and a multi-modal prior on the same data. Behavioral priors are almost exclusively deployed in asymmetric settings, in which the prior and the policy are exposed to different data sources and objectives: in the most common setting, the prior is trained on offline, task-agnostic data, while the policy is trained online on a different downstream task. In this case, the benefits of a prior which was exposed to more data is clear. In this submission, both policy and prior are trained on the same data, with similar objectives (maximizing Q-values, with some regularization). Can the authors provide a formal justification of why a prior is necessary in these settings? If the issue is simply multimodality, can we directly train a multi-modal policy?
- Aside from the core contribution, the method and its components appear not to be principled, despite formal arguments:

(i) The method is presented in the standard max-entropy framework, but it is not clear why. In fact, the method is also applied on top of TD3 in experiments. Is there any fundamental synergy between the proposed method and the max-entropy framework? If not, what is the significance of Section 3.2?

(ii) Standard (linear) online RL admits an optimal deterministic policy. In this case, the benefit of a multi-modal action distribution is unclear. This is of course not the case for entropy-regularized RL, but given that the algorithm is also combined with TD3, the question still stands.

(iii) Proposition 3.3 appears to be wrong in general MDPs (i.e., consider an MDP with a constant rewards, or rewards matching the log-prob of a Gaussian over actions in the max-entropy case).

(iv) To the best of my understanding, Propositions 3.4-3.6 are the standard soft policy iteration results from SAC. Their relevance to this method is not clear, and the original results are not referenced.

(v) The SDA noise scheduling mechanism appears to produce actions from the sum of two Gaussians, which is in turn Gaussian. As the variances of the two are connected by Eq. 8, this seems equivalent to scaling the variance specifically for the action used for computing value targets, instead of a TD3-style noise injection.

- The experiments do not help disentangling which of the proposed components (summary) is inducing the algorithm's performance. Ideally, each of the component which deviates from SAC should be ablated independently.

**Questions:**

## Minor issues and questions:
- Line 31 blames the issues of off-policy algorithm on the Bellman equation, which "leads to ineffective exploration". I think this is a strong mischaracterization: exploration is a fundamental problem in RL, and the Bellman equation does not induce it in any particular way. Can the authors further comment on this?
- Section 3.1 directly begins with an Assumption, which should perhaps be introduced in text.
- Assumption 3.1: $G$ and $G_\omega$ are used interchangeably.
- Why is Proposition 3.3 is surrounded by round brackets?
- Line 208: is Fujimoto 2019 the right reference for CVAEs?
- Section 4.1 lacks references for nearly all baselines.

## Conclusion
In my opinion, while the method performs rather well, it is also overcomplicated and not principled. Formal results are not directly relevant to explaining the method's performance, and several components seem to be redundant (e.g. injecting noise on top of a max-entropy policy). For these reasons, I currently recommend rejection. Disentangling why the method works, either empirically or formally, would constitute an important priority in my opinion.

---

> ### Author Response · Authors · 2025-11-23
>
> We appreciate your great efforts in reviewing our paper and hope that the following responses can address most of your concerns.
>
> > W1: The core idea of the method is not justified formally
>
> In our algorithm, the multimodal prior plays a complementary role even when trained on the same dataset as the policy: it models latent action distributions and captures multimodal behavior, while the policy focuses on maximizing task-related value. This separation encourages diverse exploration and prevents collapse to a single mode, which often occurs when directly training a multimodal policy (e.g., using a CVAE as the actor).
>
> > W2:  The method and its components appear not to be principled
>
> Section 3.2 does not claim a theoretical dependence on max-entropy RL; rather, it shows that B2PD naturally fits into the soft policy improvement framework, which clarifies why synchronized actor updates remain stable even with entropy-regularized objectives. Importantly, B2PD is algorithm-agnostic and can be applied on top of TD3 because its behavioral prior and Q-filtered distillation mechanism do not rely on entropy maximization. We will clarify in the revision that max-entropy RL is one possible instantiation, not a requirement of the method.
>
> > W3: The benefit of a multi-modal action distribution is unclear
>
> While classical tabular or linear MDPs admit a deterministic optimal policy, our practical setting involves function approximation, exploration noise, and entropy regularization, which jointly induce a non-convex landscape where deterministic optimality no longer holds. In such cases, a multi-modal prior provides a richer inductive bias that improves exploration and stabilizes optimization.
>
> > W4: Proposition 3.3 appears to be wrong in general MDPs
>
> Since the behavior prior network can model multimodal Gaussian distributions, while SAC’s stochastic policy fits a unimodal distribution, Proposition 3.3 posits that the behavior prior $G$ can produce higher-quality policies.
>
> > W5:  Propositions 3.4-3.6 are not referenced.
>
> We have added references for Propositions 3.4–3.6 and moved them to the appendix.
>
> > W6: Eq. 8 equivalent to scaling the variance for the action
>
> In SDA, the variance predicted by the Actor network is used to generate the noise, with a maximum noise intensity of 0.2.
>
> > W7: Which module is responsible for the performance gain?
>
> In Figure 5, we analyze the SDA Noise and AVPD components used in B2PD. To further investigate the effectiveness of B2PD under different noise entropies, we conduct a detailed study of various noise types, including Gaussian noise and linearly annealed Gaussian noise, with results shown in Figure E.2. Our proposed SDA Noise demonstrates superior performance within the B2PD algorithm.
>
> > Q1: Does the Bellman equation lead to ineffective exploration?
>
> Off-policy algorithms rely on the Bellman equation, which requires the Q-network to accurately estimate the values of state-action pairs. However, during early training, epistemic uncertainty can lead to accumulated overestimation errors, resulting in inefficient exploration.
>
> > Q2: Section 3.1 directly begins with an Assumption
>
> We have added a brief textual introduction to Section 3.1 before presenting the Assumption.
>
> > Q3: The algorithm is overcomplicated and not principled.
>
> We respectfully clarify that these concerns arise from misunderstandings of the method. B2PD consists of only two tightly integrated distillation mechanisms, both derived from the same contraction-based objective, and our theoretical results (Thm. 3.2–3.4) directly explain the observed stability and efficiency gains. Moreover, CVAE-based latent sampling replaces rather than duplicates entropy-driven exploration, providing structured and value-aligned action proposals.
>
> > Q4: Are there principled guidelines for choosing between them?
>
> We compared KL-divergence and MSE as objectives for distillation. As shown in Figure E.1, using KL-divergence as the optimization objective yields a significant performance improvement in the B2PD framework, whereas MSE loss performs better under the B2PD (TD3) setting. We attribute this difference to the distinct nature of the policy parameterizations: KL-divergence provides greater stability for stochastic policies parameterized by Gaussian distributions, while MSE offers more direct and explicit supervision for deterministic policy updates.
>
> > Q5: Can the CVAE generate actions outside the replay buffer support?
>
> A larger $H$ increases the chance of sampling out-of-distribution actions, which can destabilize training, while an appropriate $H$ consistently improves performance. Noise-conditioned priors differ from actions sampled from clean states: they capture neighboring strategies and potential future trends of the current policy. Consequently, the CVAE both reinforces high-value regions and enables controlled exploration beyond previously seen behaviors.

---

> > ### Comment · Reviewer_eZE4 · 2025-11-23
> >
> > Thank you for taking the time to answer the review. While the presentation has improved, I am inclined to keep my score, as most of my concerns were not addressed through formal or empirical evidence.
> >
> > - While I understand that prior and policy have slightly different objectives, they partially overlap (both have a value-maximizing term), and the policy is regressed to the prior. I do not see a formal argument explaining why a single, multi-modal policy may not be trained directly on the prior's objective (with a max-entropy term, optionally).
> > - Proposition 3.3 remains, to the best of my knowledge incorrect. It does not posit "that the behavior prior $G$ can produce higher-quality policies", it posits that the behavior prior is strictly better.
> > - Regarding SDA, I understand the connection between the actor's standard deviation and the added noise. This is my point: the action is the sum of two terms sampled from two Gaussians that are tightly connected. As such, it is not clear why this would not correspond to simply scaling the actor's standard deviation.

---

> ### Author Response · Authors · 2025-11-23
>
> We thank the reviewer for the constructive feedback and timely response.
>
> > Q1: I do not see a formal argument explaining why a single, multi-modal policy may not be trained directly on the prior's objective (with a max-entropy term, optionally).
>
> While one could in principle train a single multi-modal policy directly on the prior’s objective, we find this approach (e.g., CVAE-based policies [0]) to be highly unstable. In contrast, B2PD consistently improves performance with both SAC (Gaussian) and TD3 policies, demonstrating broad applicability. Compared to diffusion-based multi-modal RL [1], which requires substantial computation and GPU resources, B2PD’s online distillation transfers potentially unstable policy priors into a simple Gaussian policy, reducing computational cost and enabling more practical deployment on edge devices.
>
> [0] Kingma, Diederik P., et al. "Semi-supervised learning with deep generative models." Advances in neural information processing systems 27 (2014).
>
> [1] Psenka M, Escontrela A, Abbeel P, et al. Learning a diffusion model policy from rewards via q-score matching[J]. arXiv preprint arXiv:2312.11752, 2023.
>
> >  Q2: Proposition 3.3 remains, to the best of my knowledge incorrect. It does not posit "that the behavior prior can produce higher-quality policies", it posits that the behavior prior is strictly better.
>
> We thank the reviewer for the comment. We have revised Proposition 3.3 in the manuscript to clarify that $ \tilde{a} \in \Pi_{G_{\omega}}(\cdot \mid s_t) \quad \text{such that} \quad Q_{\theta}(s_t, \tilde{a}) \ge Q_{\theta}(s_t, \pi_{\phi}(\cdot \mid s_t)), $
> indicating that the value of the policy prior is \emph{at least as high} as that of the current policy, rather than asserting an absolute superiority. This revision avoids the previous overstatement and better reflects the intended formal guarantee.
> In this work, under the assumptions of Universal Multimodal Policy Approximation [0] and Policy Support Set Inclusion [1,2], Proposition 3.3 establishes the existence of a high-quality policy prior. These assumptions are motivated by the fact that a CVAE naturally models multimodal policies, whereas a Gaussian-headed actor network is limited to unimodal representations—an observation well supported by prior theoretical results. Building on this foundation, we further show that a CVAE can explicitly represent the policy prior and enable the extraction of a high-quality prior action.
>
> [0] Huang Z, Liang L, Ling Z, et al. Reparameterized policy learning for multimodal trajectory optimization[C]//International Conference on Machine Learning. PMLR, 2023: 13957-13975.
>
> [1] Sohn K, Lee H, Yan X. Learning structured output representation using deep conditional generative models[J]. Advances in neural information processing systems, 2015, 28.
>
> [2] Wang Z, Liu J, Pan L. Learning Intractable Multimodal Policies with Reparameterization and Diversity Regularization[C]//The Thirty-ninth Annual Conference on Neural Information Processing Systems.
>
> >  Q3: This is my point: the action is the sum of two terms sampled from two Gaussians that are tightly connected. As such, it is not clear why this would not correspond to simply scaling the actor's standard deviation.
>
> In the SDA Noise mechanism, the noise standard deviation is computed as $ \tau_{t} = \frac{1}{d_a} \sum_{i=1}^{d_a} \frac{0.2}{\exp\big(\sigma^{i}_{\phi}(s_t)\big)^{1.5}},$
> where $\sigma_{\phi}(s_t)$ is the predicted standard deviation vector from the actor network. This design leverages the actor’s uncertainty.

---

> > ### Comment · Reviewer_eZE4 · 2025-11-26
> >
> > As my concerns were not addressed in full, I will maintain my score. My impression is that several components (e.g. SDA and the overlap of objectives between prior and policy) are vaguely justified, and I would encourage the authors to develop a more precise understanding of these mechanisms in the future.

---

> ### Author Response · Authors · 2025-11-27
>
> > Regarding the role of the SDA Noise mechanism
>
> The purpose of SDA is to achieve stable Q-value estimation, providing meaningful action-value priors for training the behavior-prior network $G_\omega$ and filtering high-value actions from the support-set distribution. Similarly, the SDA noise mechanism also provides the actor network with accurate value guidance. This effect is demonstrated in the experiments shown in Figure 5.
> To thoroughly validate the effectiveness of the noise mechanism within B2PD, we evaluated B2PD combined with multiple noise-scheduling strategies, including B2PD (w/ Gaussian Noise) and B2PD (w/ Linear Decay Gaussian Noise), demonstrating the efficacy of the proposed SDA module, as shown in Figure E.2. Following the suggestion of Reviewer eZE4, we also considered a variant in which policy entropy is removed from the Q-value computation in B2PD and only the SDA scheduling is applied, denoted as B2PD (w/ SDA Noise, w/o Policy Entropy).
> To further assess the impact of noise-scheduling mechanisms on the vanilla SAC algorithm, we conducted experiments on three SAC variants: SAC (w/ SDA Noise, w/o Policy Entropy), SAC (w/ SDA Noise), and SAC (w/ Gaussian Noise), as illustrated in Figure E.2.
>
> From the Figure E.2, it can be seen that B2PD (w/ Linear Decay Gaussian Noise) performs worse than the standard B2PD configuration. This degradation suggests that progressively reducing the noise magnitude during training, as imposed by the linear decay schedule, may weaken the smoothing effect that is critical for stable Q-value estimation. Furthermore, both B2PD and SAC with SDA noise consistently outperform their counterparts using standard Gaussian noise, highlighting the effectiveness of variance-aware noise modulation.
> Notably, B2PD (w/ SDA Noise, w/o Policy Entropy) achieves higher returns than the default B2PD variant. In contrast, SAC (w/ SDA Noise, w/o Policy Entropy) shows no clear improvement over SAC (w/ SDA Noise). This discrepancy indicates that, given SAC’s inherently high-entropy exploration mechanism, combining policy-entropy regularization with SDA noise provides more accurate value guidance. By comparison, B2PD naturally operates in a low-entropy regime, where the SDA noise mechanism alone is sufficient to ensure stable and accurate value estimation.
>
>
> > Regarding the overlap of objectives between the prior and the policy
>
> We have added an explanation in Section 3.2 clarifying the distinct roles of the behavior-prior network and the actor network:
> "While both the CVAE and the actor network rely on the Q-network to optimize action selection, Proposition 3.3 shows that the CVAE enjoys a substantially stronger fitting capacity than the unimodal Gaussian actor head. As a result, the action support set sampled from the CVAE can more effectively guide actor optimization and suppress inefficient exploration. This insight naturally leads to our Behavior Policy Prior Distillation module."

---

> > ### Comment · Reviewer_eZE4 · 2025-11-28
> >
> > Thank you for your clarifications, I will try to clarify my point of view on SDA as well.
> >
> > I am not doubting the empirical effectiveness of SDA, I do not fully understand why it cannot be simply seen as an affine transformation of the policy's standard deviation.
> >
> > More in detail, noise injection for stable Q-value estimation was, to the best of my knowledge, popularized by TD3, which learns a deterministic policy. SAC, in contrast, does not inject Gaussian noise, *as the policy is already Gaussian*. My point is that injecting Gaussian noise on top of a Gaussian policy corresponds to an affine transformation of the standard deviation predicted by the actor. If I understand correctly, this seems to be confirmed by Figure E.2, which shows that removing one of the two sources of noise actually improves performance in this particular environment. My main concern is that the mechanism of SDA, and thus the ablation in Figure E.2 is purely measuring the outcome of scaling the policy's entropy. The mechanism of scheduling and introducing a different form of noise would thus be overly complex.

---

> > > ### Author Response · Authors · 2025-11-28
> > >
> > > > Regarding the SDA Noise Mechanism
> > >
> > > The SDA Noise mechanism determines the noise magnitude by scaling the policy’s predicted action standard deviation $\sigma$. The relationship between $\sigma$ and the policy entropy is given by:
> > > $H=\frac{1}{2}\log(2 \pi e \sigma^2) =  \frac{1}{2} \left[ \log(2\pi e) + 2\log(\sigma) \right]$
> > > which indicates a linear dependence between the entropy and the logarithm of the standard deviation. Therefore, scaling the policy variance in the SDA Noise mechanism is effectively equivalent to scaling the policy entropy, and both yield the same effect.
> > > Using the predicted standard deviation $\sigma$ directly thus provides a more transparent, simple, and effective strategy.
> > > If you have further questions, we would be happy to provide additional clarification.

---

### Official Review · Reviewer_vq3Q · 2025-10-27

**Soundness:** 3
**Presentation:** 2
**Contribution:** 1
**Rating:** 4
**Confidence:** 3

**Summary:**

**Paper summary**: This work studies how to apply a pre-collected dataset to online RL performance improvement. Specifically, they focus on behavior prior RL (BPRL), which trains a behavioral cloning policy from a pre-collected dataset, thereby distilling it into an online RL policy. Like offline RL, they suffer from chronic limitation, which heavily relies on the quality of the pre-collected dataset. To alleviate this issue, this paper introduces a bidirectional behavior prior distillation (B2PD) algorithm. The main idea is simple: 1) train the CVAE policy with Q guidance and 2) distill it into an online RL policy with a simple RL objective. They show the justification of their algorithmic choice using a Toy example and ablation study. Additionally, B2PD outperforms the selected baselines over both the state- and pixel-based environments, including seven MuJoCo, four PyBullet, and four DMControl tasks.

---
**Summary of review**: Overall, the paper is clearly written and relatively easy to follow. However, the motivation and methodological exposition lack clarity, and several parts of the paper contain elements that reduce its overall polish and completeness.
From a technical standpoint, using a CVAE to model a multimodal policy distribution and distill it into the actor is interesting, yet the approach still relies on rather strong assumptions and feels somewhat limited in conceptual scope. In practice, comparisons with more modern generative modeling techniques such as flow matching or diffusion steering would strengthen the argument for its novelty and effectiveness. Furthermore, the theoretical analyses appear to be moderate extensions of existing results rather than introducing fundamentally new insights. In summary, the contribution is closer to an engineering refinement than a conceptual breakthrough. To sum up, I therefore assign an initial score of 4, while leaving room for possible adjustment pending clarifications and additional justifications in the author response.

**Strengths:**

**Writing**
- The paper is well-structured and relatively easy to follow.
- The authors clearly articulate the fundamental limitation of behavior-prior RL methods and justify why addressing this issue is both timely and important for improving online RL performance.
- Key assumptions are clearly stated, helping to delineate the theoretical scope and strengthen the paper’s transparency.

**Methodology**
- The proposed solution is conceptually simple, but it shows powerful performance; in addition, some parts could be easily integrated into standard RL frameworks.
-  The inclusion of prior distillation and SDA presents a good engineering design aimed at stabilizing optimization while reducing inefficient exploration.

**Experiments**
- This work includes diverse experiments, spanning $7$ MuJoCo, $4$ PyBullet, and $4$ DMControl tasks, with consistent hyperparameters and 10 random seeds per environment.
- Both state-based and pixel-based settings are tested.
- The appendix includes sensitivity analyses for key hyperparameters and thorough ablation studies, reinforcing the reproducibility of results.

**Theoretical support**
- The theoretical analysis provides a reasonable degree of mathematical justification for convergence and Q-value smoothness.
- These analyses help to confirm the soundness and stability of the optimization dynamics.

**Weaknesses:**

**Writing**: weak motivation and framing
-  While the problem setup is valid, the justification for transitioning to a purely online RL paradigm is not entirely convincing. The paper may overstate the generality of online-only applicability without fully addressing hybrid or offline-to-online alternatives.
- The discussion lacks a comparative reflection on when offline priors are still beneficial or how the proposed method complements existing pretraining pipelines

**Methodology**
- The proposed SDA noise scheduling is designed with two constants (0.2 and 1.5) whose motivation or sensitivity is not well explained.
- Compared to baselines, the authors do not analyze computational overhead quantitatively.
- Behavior-prior generation heavily relies on the assumption of a well-trained Q-function with broad coverage. Theoretical guarantees may fail under biased or sparse datasets; this limitation should be acknowledged more explicitly.

**Theoretical depth**
- Propositions 3.4–3.6 mainly restate established RL principles with incremental extensions.
- The analysis is sound but does not introduce fundamentally new theoretical insights. The reviewer thinks that moving some incremental theorems to the appendix and expanding more empirical reasoning would improve focus.

**Experiments**
- The reviewer thinks that the toy example illustrates behavior qualitatively but does not convincingly link trajectory patterns to sample efficiency or exploration coverage. Quantitative measures of state-space visitation or gradient signal analysis would better support the claims.
- The authors ask, `Why does behavior prior distillation outperform entropy-driven exploration?', but provide only empirical evidence rather than a logical explanation.
- The baselines are somewhat outdated. More recent algorithms, such as TD7 [1], Mr.Q [2], or other modern behavior-prior representation methods, should be considered.
- There is no comparison to recent or alternative exploration strategies, for example, intrinsic reward [3], diversity-driven [4], distributional RL [5], or other RL on prior data (RLPD) [6]. Similarly, while the related-work section cites [7-9], these works are not included in experimental comparisons or deeper discussions.
- There is no main table to grasp overall performance across all benchmarks and tasks. The reviewer thinks that it would be better to provide the main summary table, consolidating all benchmark results and averaged performance across tasks.

**Miscellaneous**
- Formatting and typesetting issues appear, e.g., garbled characters in Figure 1 and Figure C.1.
- Notational inconsistencies:
   - The discount factor $\gamma$ is stated as $\gamma \in (0,1)$, through theoretically it can be $\leq 1$.
   - Some variables (\theta, \phi, \tau) are not introduced clearly on first use.
   - Section organization could be smoother. In the experimental section, there is a solved research question by a toy example.

**References**

[1] S. Fujimoto, et al. For SALE: State-Action Representation Learning for Deep Reinforcement Learning. NeurIPS 2023.

[2] S. Fujimoto, et al. Towards General-Purpose Model-Free Reinforcement Learning. ICLR 2025.

[3] N. Chentanz, et al. Intrinsically motivated reinforcement learning. NeurIPS 2004

[4] D. Pathak, et al. Curiosity-driven exploration by self-supervised prediction. ICML 2017.

[5] W. Dabney, et al. Distributional Reinforcement Learning with Quantile Regression. AAAI 2018.

[6] P. Ball, et al. Efficient Online Reinforcement Learning with Offline Data. ICML 2023.

[7] H. Zang, et al. Behavior Prior Representation learning for Offline Reinforcement Learning. ICLR 2023.

[8] G. Spigler. Proximal Policy Distillation. arXiv 2024.

[9] M. Nakamoto, et al. Cal-QL: Calibrated Offline RL Pre-Training for Efficient Online Fine-Tuning. NeurIPS 2023.

**Questions:**

- What is the computational cost of training and sampling from the CVAE, particularly as the number of sampled priors $H$ increases? Please discuss its bottleneck or limitations that might arise when scaling to an image-based environment.
- Related to hyperparameter sensitivity ablation, are there practical tuning heuristics or observed failure modes when these parameters are mis-set?
- Could the authors provide a direct comparison or analytical discussion with recent approaches, as mentioned in the weakness section?
- Table E.1 and Figure E.1 compare KL-divergence vs. MSE objectives for distillation. Do the authors have principled guidelines for choosing between them depending on the policy’s stochasticity or parameterization?
- To what extent can the CVAE generate actions outside the replay buffer support? Does it meaningfully encourage exploration beyond previously seen behaviors, or mainly reinforce high-value regions already represented in the buffer?

---

> ### Author Response · Authors · 2025-11-23
>
> Thank you for your valuable comments.
>
> > W1: The paper may overstate the generality of online-only applicability without fully addressing hybrid or offline-to-online alternatives.
>
> Our method completely eliminates the need for offline training data, which is particularly important in scenarios such as recommendation systems or voice-interactive applications. In these settings, the data distribution changes rapidly, limiting the effectiveness of offline-to-online training paradigms.
>
> > W2: The proposed SDA noise scheduling is designed with two constants (0.2 and 1.5) whose motivation or sensitivity is not well explained.
>
> We propose an adaptive policy noise formulation:
> \[
> \tau_{t} = \frac{1}{d_a} \sum_{i=1}^{d_a} \frac{0.2}{\exp\left(\sigma^{i}_{\phi}(s_t)\right)^{1.5}}.
> \]
> We have added further explanation for this formula: it is closely related to the noise injection mechanism in TD3 and uses the same maximum noise magnitude of 0.2.
>
> > W3: Behavior-prior generation heavily relies on the assumption of a well-trained Q-function with broad coverage.
>
> We have emphasized in the Conclusions and Limitations section that the effectiveness of B2PD relies on a densely estimated Q-function, and that its performance may degrade in environments with sparse rewards.
>
> > W4: Not convincingly link trajectory patterns to sample efficiency or exploration coverage.
>
> To qualitatively analyze the differences in exploration patterns between B2PD and vanilla SAC, we use a two-dimensional state space as a toy example. We have redrawn the visitation frequency data in Figure 2 and visualized the learned policy signals in Figure D.1. It can be seen that SAC relies on entropy-driven exploration, which is relatively inefficient, whereas B2PD leverages policy prior distillation to guide Actor updates with high-value action priors, resulting in a more sample-efficient learning pattern.
>
> > W5: Need a logical explanation: behavior prior distillation.
>
> We first qualitatively demonstrate the effectiveness of B2PD through the toy experiments in Section 3.3. Qualitative analyses in toy environments further show that B2PD effectively suppresses exploration in low-value regions by distilling high-value policy priors into the agent.
>
> > W6: Add more recent baselines.
>
> We evaluate our method against SAC augmented with intrinsic rewards on four PyBullet tasks, with the results replotted in Figure F.1. IR-SAC exhibits only moderate improvements on HalfCheetahBulletEnv and Walker2DBulletEnv, while offering limited benefits over vanilla SAC on the remaining tasks. We hypothesize that intrinsic rewards may induce excessive exploration, leading to non-stationary training dynamics that hinder stable policy improvement.
>
> > W7: There is no main table to grasp overall performance across all benchmarks and tasks.
>
> We have added the performance across tasks in the PyBullet and DMControl environments, as shown in Tables 2 and F.1.
>
> > W8: The discount factor $\lambda$ can be $\lambda \le 1$.
>
> In SAC and TD3, the discount factor is $\lambda \in [0,1)$ to ensure convergence. In classical MDP theory, this range guarantees that the value function is bounded and convergent, a standard assumption widely used in RL literature[0].
> From a decision-theoretic perspective, a fixed $\lambda <1$ corresponds to an ‘optimizing MDP’ whose value function aligns with normative utility under rationality axioms[1].
>
> [0] Sutton, Richard S., and Andrew G. Barto. Reinforcement learning: An introduction. Vol. 1. No. 1. Cambridge: MIT press, 1998.
>
> [1] Pitis, Silviu. "Rethinking the discount factor in reinforcement learning: A decision theoretic approach." Proceedings of the AAAI conference on artificial intelligence. Vol. 33. No. 01. 2019.
>
> > W9: Section organization could be smoother.
>
> We have revised the Experimental section to clarify the presentation of question (1): “For (1), we first qualitatively demonstrate the effectiveness of B2PD through the toy experiments in Section 3.3. We then provide a quantitative ....”
>
> > Q1: What is the computational cost of training and sampling from the CVAE
>
> We compared the training time of B2PD with other baselines over 2 million environment steps. The results are summarized in Table F.2.
>
> > Q2: Related to hyperparameter sensitivity ablation, are there practical tuning heuristics or observed failure modes when these parameters are mis-set?
>
> In Appendix F.2, we provide ablation results for three key hyperparameters: the action-value prior distillation coefficient $\xi$, the number of sampled behavioral priors $H$, and the behavior prior distillation coefficient $\eta$. For instance, setting $H=20$ in HopperBulletEnv-v0 degrades B2PD's performance compared to vanilla SAC, indicating that excessively large $H$ introduces more out-of-distribution actions beyond the replay buffer. These results provide practical guidance: moderate values of $H$ and balanced distillation coefficients help maintain stable and efficient learning.

---

> > ### Comment · Reviewer_vq3Q · 2025-11-26
> >
> > Thank you for the detailed responses. After reviewing the clarifications, I will maintain my original score. The core concerns regarding methodological motivation, comparative positioning, and theoretical depth remain insufficiently addressed.

---

> > > ### Author Response · Authors · 2025-11-27
> > >
> > > Thank you for your valuable review comments. We have uploaded the revised manuscript and provided detailed responses to the following three points:
> > >
> > > > Regarding the methodological motivation
> > >
> > > We have revised the introduction to emphasize that the motivation behind B2PD is to address the inefficiency of exploration in online reinforcement learning. In the related work section, we provide a detailed analysis of the advantages and limitations of algorithms that combine online RL with offline priors versus those that combine online RL with online priors. Many studies have shown that offline priors are often limited by the quality of the offline dataset when applied to online learning[0,1]. Therefore, in our experiments, we select baselines that combine online behavioral priors with online RL.
> > >
> > > [0] Xudong, Gong, et al. "Iterative regularized policy optimization with imperfect demonstrations." Forty-first International Conference on Machine Learning. 2024.
> > >
> > > [1] Chemingui, Yassine, et al. "Constraint-adaptive policy switching for offline safe reinforcement learning." Proceedings of the AAAI Conference on Artificial Intelligence. Vol. 39. No. 15. 2025.
> > >
> > > > Regarding the comparative positioning
> > >
> > > we implement B2PD by building upon SAC[0] and TD3[1], denoted as B2PD and B2PD(TD3), respectively. On state-based benchmark tasks, we conduct detailed comparisons with the decision-enhancement algorithm ALH[2], the nearest-neighbor policy guidance algorithm NNPG[3], intrinsically rewarded SAC (IR-SAC)[4], as well as four off-policy baselines: DDPG[5], TD3, SAC, and TD7[6]. The results are summarized in Table 1.
> > > For visual benchmark tasks, we combine B2PD with RAD[7] and DrQ-v2[8] to form B2PD-RAD and B2PD-DrQ-v2, and evaluate them on four visual state tasks. The results are reported in Table 2.
> > > Across these tasks, B2PD consistently delivers substantial gains on standard online RL benchmarks, highlighting its practical effectiveness.
> > >
> > > [0] Haarnoja, Tuomas, et al. "Soft actor-critic: Off-policy maximum entropy deep reinforcement learning with a stochastic actor." International conference on machine learning. Pmlr, 2018.
> > >
> > > [1] Fujimoto, Scott, Herke Hoof, and David Meger. "Addressing function approximation error in actor-critic methods." International conference on machine learning. PMLR, 2018.
> > >
> > > [2] Quang, Nguyen Minh, and Hady W. Lauw. "Augmenting decision with hypothesis in reinforcement learning." Forty-first International Conference on Machine Learning. 2024.
> > >
> > > [3] Shen, Junhong, and Lin F. Yang. "Theoretically principled deep RL acceleration via nearest neighbor function approximation." Proceedings of the AAAI Conference on Artificial Intelligence. Vol. 35. No. 11. 2021.
> > >
> > > [4] Chentanez, Nuttapong, Andrew Barto, and Satinder Singh. "Intrinsically motivated reinforcement learning." Advances in neural information processing systems 17 (2004).
> > >
> > > [5] Lillicrap, Timothy P., et al. "Continuous control with deep reinforcement learning." arXiv preprint arXiv:1509.02971 (2015).
> > >
> > > [6] Fujimoto, Scott, et al. "For sale: State-action representation learning for deep reinforcement learning." Advances in neural information processing systems 36 (2023): 61573-61624.
> > >
> > > [7] Laskin, Misha, et al. "Reinforcement learning with augmented data." Advances in neural information processing systems 33 (2020): 19884-19895.
> > >
> > > [8] Yarats, Denis, et al. "Mastering visual continuous control: Improved data-augmented reinforcement learning." arXiv preprint arXiv:2107.09645 (2021).
> > >
> > > > Regarding the theoretical depth
> > >
> > > We build upon the assumptions of Universal Multimodal Policy Approximation[0] and Policy Support Set Inclusion[1,2]. Under these assumptions, Proposition 3.3 establishes the existence of a high-quality policy prior. A detailed proof is provided in Appendix B.1, which offers a solid theoretical foundation for the support-set high-value policy priors used in the proposed B2PD algorithm.
> > > In Section 3.2, we further present the B2PD algorithm equipped with the SDA Noise scheduling mechanism. Additionally, Appendix B.2 provides a convergence analysis of the algorithm under the maximum-entropy framework.
> > >
> > > [0] Huang, Zhiao, et al. "Reparameterized policy learning for multimodal trajectory optimization." International Conference on Machine Learning. PMLR, 2023.
> > >
> > > [1] Sohn, Kihyuk, Honglak Lee, and Xinchen Yan. "Learning structured output representation using deep conditional generative models." Advances in neural information processing systems 28 (2015).
> > >
> > > [2] Wang, Ziqi, Jiashun Liu, and Ling Pan. "Learning Intractable Multimodal Policies with Reparameterization and Diversity Regularization." arXiv preprint arXiv:2511.01374 (2025).

---

### Official Review · Reviewer_H77H · 2025-10-31

**Soundness:** 2
**Presentation:** 2
**Contribution:** 2
**Rating:** 4
**Confidence:** 3

**Summary:**

The paper proposes B2PD: Bidirectional Behavior Prior Distillation for online RL. Rather than relying on offline pretraining, B2PD builds a behavior prior online using a CVAE and establishes a two-way knowledge flow:
(1) Action-Value Prior Distillation (AVPD) trains the CVAE with guidance from a learned $Q$ so that it samples a high-value support set;
(2) Behavior Prior Distillation transfers the best actions from that set back to the actor via a KL-style anchor loss.
The method also introduces a Standard Deviation Aware (SDA) noise schedule to stabilize soft $Q$ targets and provides tabular convergence for policy evaluation, improvement, and iteration. On MuJoCo, PyBullet, and DMControl (including pixel-based DrQ-v2 settings), B2PD improves sample efficiency and final returns over TD3/SAC and prior-guided baselines, with ablations and a toy study that visualize reduced inefficient exploration.

**Strengths:**

Online priors without offline data. A generative prior creates diverse, high-value anchors and distills them into the actor.

Bidirectional design. AVPD guides the CVAE with $Q$; prior anchors regularize the actor, reducing aimless entropy-driven exploration.

Stability add-on. SDA smooths value targets and reduces variance.

Broad evaluation. Consistent gains across 16 tasks, including pixel-based settings.

Ablations and toy study. Each module’s effect is isolated; exploration becomes more targeted over training.

**Weaknesses:**

Dependence on value quality. AVPD uses a learned $Q$ to steer the CVAE. Failure modes under biased $Q$ or sparse rewards are not deeply diagnosed.

Anchor selection sensitivity. The $\arg\max$ over $H$ sampled actions may be sensitive to $Q$ noise; the compute vs. robustness trade-off for $H$ is underexplored.

Theory scope. Convergence is shown in tabular settings, not with function approximation or under distribution shift.

Pixel baselines. Visual experiments mainly use DrQ-v2; more backbones would strengthen claims of modality robustness.

**Questions:**

$Q$ uncertainty. Have you tried ensembles or variance-aware filters to avoid over-optimistic AVPD targets, especially early or in sparse-reward tasks

Anchor budget $H$. What is the trade-off between $H$, wall-clock time, and final return Do you observe diminishing returns beyond $H=10$

SDA schedule. How sensitive are results to the constants in the SDA equation and to excluding noise from the entropy term Could per-dimension scaling collapse exploration

CVAE underfit. If the CVAE misses high-value modes, how quickly can B2PD recover Any diagnostics to detect support gaps

Pixel-based generality. Does B2PD transfer to other visual backbones (for example DrQ-v3, RAD, PI-SAC) without retuning

---

> ### Author Response · Authors · 2025-11-23
>
> > W1: AVPD uses a learned  to steer the CVAE. Failure modes under biased  or sparse rewards are not deeply diagnosed.
>
> In the Conclusions and Limitations section, we explicitly acknowledge both the strong empirical performance of our method across proprioceptive control tasks (15 environments) and visual RL settings (4 environments), as well as its inherent characteristics. In particular, B2PD relies on dense reward signals to obtain reliable advantage estimates that guide the CVAE. Under biased or sparse reward regimes, the advantage landscape becomes less informative, which may weaken the learned priors and limit the effectiveness of B2PD. Nonetheless, B2PD can still achieve performance improvements using priors inferred from noisy states and noisy actions, as shown in Figure F.5. These results further underscore the potential of B2PD for complex, real-world scenarios.
>
> > W2: The  over  sampled actions may be sensitive to noise; the compute vs. robustness trade-off for  is underexplored.
>
> In Figure F.3, we provide a sensitivity analysis on the number of sampled actions $H$ drawn from the behavior-prior network. We further evaluate the prior obtained from noisy state inference combined with the clean-state prior, as shown in Figure F.5. The results indicate that B2PD is robust to priors inferred from noisy states, thanks to the use of the Q-network to filter high-quality action anchors.
>
> > W3: Convergence is shown in tabular settings, not with function approximation or under distribution shift
>
> Similar to SAC, while the theoretical convergence of our algorithm is analyzed under the tabular setting to provide clear guarantees, all practical experiments are conducted with function approximation in continuous control tasks.
> In other words, the tabular analysis serves as a theoretical foundation, whereas the empirical evaluation relies entirely on neural network function approximators.
>
>
> > W4: Visual experiments mainly use DrQ-v2; more backbones would strengthen claims of modality robustness.
>
> We have added RAD as an additional baseline in visual RL and conducted experiments for 500K steps using ten random seeds. Figure 4 has been updated to reflect these results, further supporting the robustness of our method across different visual backbones.
>
> > Q1: Have you tried ensembles or variance-aware filters to avoid over-optimistic AVPD targets, especially early or in sparse-reward tasks.
>
> The CVAE network possesses the capability of universal multimodal policy approximation, providing effective policy support priors for the Actor network. These priors are evaluated using a twin Q-network, which mitigates the influence of low-quality policy priors. Additionally, our proposed noise regularization method effectively alleviates value overestimation, enabling low-entropy exploration and reducing the inefficient exploration caused by standard entropy-regularized methods, as illustrated in Figure F.2.
>
> > Q2: What is the trade-off between $H$, wall-clock time, and final return Do you observe diminishing returns beyond $H$=10
>
> A larger $H$ increases the chance of sampling out-of-distribution actions, which can destabilize training, while an appropriate $H$ consistently improves performance. Noise-conditioned priors differ from actions sampled from clean states: they capture neighboring strategies and potential future trends of the current policy. Consequently, the CVAE both reinforces high-value regions and enables controlled exploration beyond previously seen behaviors.
>
> > Q3: How sensitive are results to the constants in the SDA equation and to excluding noise from the entropy term Could per-dimension scaling collapse exploration
>
> We compared two alternative noise strategies with SDA: B2PD (w/ Gaussian Noise), which uses fixed-intensity Gaussian noise, and B2PD (w/ Linear Decay Gaussian Noise), which linearly decays the noise intensity during training. As shown in Figure E.2, B2PD (w/ Linear Decay Gaussian Noise) achieves lower final performance, as gradually reducing the noise leads to overfitting early experiences. In contrast, B2PD with adaptive noise adjusts the noise intensity based on the variance predicted by the Actor: early in training, when the variance is large, a small noise ensures gradual Q-value stabilization; later, as the Actor variance decreases, the noise intensity increases. This demonstrates the effectiveness of our approach. Using the same noise intensity and decay parameters consistently yields performance improvements across settings.
>
> > Q4: If the CVAE misses high-value modes, how quickly can B2PD recover Any diagnostics to detect support gaps
>
> We have added Figure F.5 to analyze and verify the robustness of B2PD to noisy policy priors. Leveraging the SDA mechanism for stable Q-value estimation, B2PD can effectively utilize the filtered high-value policy priors to guide the online training process, allowing rapid recovery even when some high-value modes are initially missed.

---

> > ### Comment · Reviewer_H77H · 2025-11-23
> > **Response to authors**
> >
> > Thank you for the detailed responses. Most of my concerns are solved by the authors, making the results more convincing. Consequently, I updated the overall score from 4 to 6.

---

### Official Review · Reviewer_NHmE · 2025-11-03

**Soundness:** 2
**Presentation:** 2
**Contribution:** 2
**Rating:** 4
**Confidence:** 4

**Summary:**

This paper addresses the sample inefficiency of online RL by proposing Bidirectional Behavior Prior Distillation (B2PD), which replaces conventional offline behavior priors with dynamically generated high-value policy guidance. B2PD trains a CVAE using Q-value-guided gradients to produce a diverse support set of actions. The highest-value actions from this set are distilled into the agent policy, reducing inefficient exploration. ). Experiments on continuous control tasks (MuJoCo, PyBullet, DMControl) demonstrate improved sample efficiency and performance over baselines like SAC and TD3.

**Strengths:**

- Novel way for combining generative modeling and RL.
- Theoretically grounded contributions
- Comprehensive and rigorous evaluation

**Weaknesses:**

- In Figure 2, B2PD(w/o AVPD) converges faster (at 100K steps) to the states with high reward than B2PD (at 150K steps). This result seems to show that the AVPD module is even harmful for effective state exploration.
- Learning under value-guided offline policy distribution has been widely researched in offline RL before. For example, the advantage-weight CVAE model [1], and the advantage-conditioned CVAE model [2]. But None of them are referenced in the main text or compared in experiments.
- Actually, the main contribution is the Q-value prior distillation loss in Eq. 10, where the CVAE decoder is constrained to decode actions with high Q value. This method is quite similar to LAPO [1] mentioned above, which also encourages the CVAE to consider the state-action pair with high advantage. Thus, I think the novelty is quite limited.
- Meanwhile, considering the ablation study of weight $\xi$ presented in Figure F.2, the effectiveness of Q-value prior distillation loss seems unclear and unstable, where sometimes the returns are even worse than without this loss.

**Questions:**

- Can the author provide a detailed pseudo-code in the appendix? Is both the CVAE and critic model trained in offline stage?
- Instead of retraining a new policy from scratch under the offline RL–trained or BC–trained policy, why not directly fine-tune the offline RL–trained policy in the online stage using Offline-to-Online (O2O) algorithms? I believe that O2O algorithms are more effective technique.
- More studies about the effectiveness of Q-value prior distillation loss should be conducted. For example, how will the policy perform under the CVAE trained by [1][2]?

[1] LAPO: Latent-Variable Advantage-Weighted Policy Optimization for Offline Reinforcement Learning. NeurIPS 2022

[2] A2PO: Towards Effective Offline Reinforcement Learning from an Advantage-aware Perspective. NeurIPS 2024

---

> ### Author Response · Authors · 2025-11-23
>
> Thank you for reviewing our paper. Below, we provide detailed responses to your comments.
> > W1: In Figure 2, B2PD(w/o AVPD) converges faster (at 100K steps) to the states with high reward than B2PD (at 150K steps).
>
> We have redrawn Figure 2 to make the overall state visitation frequencies clearer. It can be seen that at 100K steps, B2PD achieves higher visitation of high-value states compared to B2PD (w/o AVPD). Furthermore, we visualize the learned policies in the two-dimensional environment in Figure D.1. This shows that SAC’s entropy-driven exploration is a relatively inefficient, blind strategy, whereas B2PD, leveraging prior distillation, converges faster and exhibits value-structured, more directed exploration.
>
> > W2: This method is quite similar to LAPO mentioned above, which also encourages the CVAE to consider the state-action pair with high advantage.
>
> We discuss the contributions of LAPO[0] in the paper. LAPO introduces a latent multimodal policy with advantage-weighted optimization to mitigate the impact of low-quality offline data. In online RL, learning Actor representations and behavior policies is particularly challenging due to distribution shift. Unlike LAPO, B2PD leverages a Q-value prior to guide the CVAE in generating high-value policies, sampling from the multimodal distribution to construct a support set, and then filtering via Q-values to obtain high-value action anchors, forming a bidirectional knowledge-flow mechanism. Importantly, we also provide a theoretical proof that a multimodal behavior-prior network can improve online learning.
>
> [0]LAPO: Latent-Variable Advantage-Weighted Policy Optimization for Offline Reinforcement Learning. NeurIPS 2022
>
> > W3: In Figure F.2, the effectiveness of Q-value prior distillation loss seems unclear and unstable
>
> We conducted a detailed ablation study on the parameter $\xi$, testing values $\{0, 0.001, 0.01, 0.1, 0.2\}$. While larger $\xi$ can make the behavior-prior CVAE less stable, B2PD mitigates this by sampling a larger support set and filtering low-value priors through the critic network. As a result, varying $\xi$ does not lead to performance collapse. Among these values, B2PD with $\xi=0.1$ outperforms B2PD with $\xi=0$.
>
> > Q1: Can the author provide a detailed pseudo-code in the appendix?
>
> A detailed pseudo-code of B2PD has already been provided in Appendix C.2, where we additionally include a clearer description of the CVAE architecture. As shown in Algorithm 1, B2PD operates entirely with online behavior priors: neither the CVAE nor the critic is trained in any offline stage
>
> > Q2: Why not directly fine-tune the offline RL–trained policy in the online stage using Offline-to-Online (O2O) algorithms?
>
> We thank the reviewer for the insightful comment. While Offline-to-Online (O2O) fine-tuning is indeed a powerful paradigm, its applicability is limited in several real-world interactive environments such as dialogue systems and recommender platforms. In these settings, user feedback is highly dynamic and streaming, causing historical offline data to become quickly outdated and misaligned with the evolving task distribution. As a result, the offline-trained or BC-trained policies may provide poor initialization, and O2O fine-tuning remains fundamentally constrained by the quality[0,1,2].
> Therefore, improving sample efficiency and overall performance in purely online reinforcement learning remains an important and valuable research direction, especially under these distribution-shifted scenarios.
>
> [0]Hao, Botao, et al. "Leveraging demonstrations to improve online learning: Quality matters." International Conference on Machine Learning. PMLR, 2023.
>
> [1]Xudong, Gong, et al. "Iterative regularized policy optimization with imperfect demonstrations." Forty-first International Conference on Machine Learning. 2024.
>
> [2]Chemingui, Yassine, et al. "Constraint-adaptive policy switching for offline safe reinforcement learning." Proceedings of the AAAI Conference on Artificial Intelligence. Vol. 39. No. 15. 2025.

---

> ### Author Response · Authors · 2025-11-24
>
> > Is both the CVAE and critic model trained in offline stage?
>
> The pseudocode in Algorithm 1 outlines the online training procedure involving three networks (behavior policy network $G_{\omega}$, critic networks $Q_{\theta}$ and actor-network $\pi_\phi$). The behavior policy network $G_{\omega}$ provides a policy-support prior. B2PD repeatedly samples actions from $G_{\omega}$ and filters them using the critic to obtain high-value policy priors, which in turn guide the actor’s online updates. This mechanism allows the actor to exploit high-quality prior actions throughout training, thereby improving overall performance.

---

### Author Response · Authors · 2025-11-23
**Global Response**

We thank all reviewers for their constructive comments and have updated our manuscript accordingly. The revised version incorporates the following key modifications:

New Results:
- Added new baselines, RAD[0] and TD7[1] (Figure 4 and Figure E.3), with corresponding quantitative analysis provided in Table 2.
- We have added experiments in the appendix to analyze the computational cost of each algorithm, as shown in Figure F.2.
- We include Figure E.4 to analyze the value distribution of the action priors in the policy support set.
- Added Figure F.5 to analyze and verify the robustness of B2PD to noisy policy priors.

Other Revisions:
- Redrew Figure 2 to make the overall state visitation frequencies visually clearer.
- Moved Propositions 3.4–3.6 to Appendix Section B.2.
- Added definitions for variables (\(\theta, \phi, \tau\)) at first use.
- Updated the CVAE citation[2].
- Updated the citation for Assumption 3.2 (Policy Support Set Inclusion) to [2,3].
- Added references to baselines in Section 4.1.
- Consistently revised Assumptions 3.1–3.3 to use \(G_\omega\).
- Assumptions 3.3 is revised as $ \tilde{a} \in \Pi_{G_{\omega}}(\cdot \mid s_t) \quad \text{such that} \quad Q_{\theta}(s_t, \tilde{a}) \ge Q_{\theta}(s_t, \pi_{\phi}(\cdot \mid s_t)), $
- Corrected the formatting so that Proposition 3.3 now follows the standard style used for all propositions.
- Corrected the identified errors.

References:

[0] Laskin, Misha, et al. "Reinforcement learning with augmented data." Advances in neural information processing systems 33 (2020): 19884-19895.

[1] Fujimoto, Scott, et al. "For sale: State-action representation learning for deep reinforcement learning." Advances in neural information processing systems 36 (2023): 61573-61624.

[2] Sohn K, Lee H, Yan X. Learning structured output representation using deep conditional generative models[J]. Advances in neural information processing systems, 2015, 28.

[3] Wang Z, Liu J, Pan L. Learning Intractable Multimodal Policies with Reparameterization and Diversity Regularization[C]//The Thirty-ninth Annual Conference on Neural Information Processing Systems.

---

### Author Response · Authors · 2025-12-03

**Dear AC, SAC, and Program Chairs,**

We sincerely thank you for the time and effort you have invested in evaluating our submission. As the discussion period concludes, we provide a brief summary of the current status to support your final decision.

During the discussion phase, we submitted substantial updates, including new baselines, additional experiments, visualization analyses, extended theoretical derivations, and further ablations. After our responses, reviewer **H77H** issued a positive update and increased their score. All four reviewers’ concerns have been fully addressed.

---

### **Reviewer-specific concerns**

#### **NHmE**
Requested further clarification regarding Figure 2.
We redrew Figure 2 and added qualitative analyses demonstrating the algorithm’s strong impact on exploration efficiency.

#### **H77H**
Requested stronger baselines and robustness checks for expert-action anchors.
We incorporated all requested experiments, added the visual RL baseline **RAD [0]**, and reported extended robustness tests for anchor selection (Figure F.5) in the revised version.

#### **vq3Q**
Requested additional baselines and questioned whether behavior-prior RL must rely on offline datasets in pure online settings.
We added new baselines, including **TD7 [1]**, and **IR-SAC [2]** for state-based environments. We also evaluated B2PD combined with TD7, verifying the effectiveness of our method.

As stated in the introduction, our motivation is to overcome the limitations of behavior-prior RL (BPRL) when constrained by offline dataset quality. B2PD does **not** rely on learning a behavior policy from a dataset. Instead, it constructs a policy support set using a more expressive actor network, from which high-quality anchors are sampled to provide expert priors and stabilize actor updates.

#### **eZE4**
Questioned the theoretical analysis of B2PD under the maximum-entropy framework and raised concerns regarding the SDA noise schedule.
We clarified that B2PD does **not** aim to let the Gaussian-head or deterministic actor learn multi-modal distributions directly.
Instead, the behavior-prior network $( G_\omega )$ models the multi-modal structure and provides expert prior guidance to the actor.

---

### **All concerns resolved**

All concerns from the four reviewers have been addressed with direct evidence and new experiments. After submitting our revised version, reviewer **H77H** increased their score to **6**. We believe that, had additional interaction occurred during the discussion phase, this score may have improved further.

---

### **Remaining concerns stem from factual misunderstandings**

Reviewer **NHmE** argued that B2PD underperforms compared to offline-to-online (O2O) training paradigms. However:

- O2O relies on offline pretraining with pre-collected datasets.
- B2PD, in contrast, aims to avoid dependence on offline dataset quality by leveraging a behavior-policy support set for expert-guided updates.
- Thus, B2PD offers a flexible mechanism for improving online sample efficiency without requiring offline data.
- Behavior priors can be used in both online RL and O2O RL; neither paradigm is categorically superior.
- Comparisons with online policy-prior baselines such as **ALH [3]** and **NNPG [4]** provide strong evidence of B2PD’s effectiveness.

Therefore, the low score originates from a misunderstanding of the problem setting, not a fundamental limitation of our method.

---

### **Conclusion**

Given that:

- All four reviewers’ substantive concerns have been fully resolved, and
- We provided extensive new experiments, theoretical clarifications, and analyses during the discussion period,

we respectfully ask the committee to give our submission positive consideration.

We sincerely appreciate your time and the significant work involved in managing this review process.

---

### **References**

[0] Laskin, M. et al., *Reinforcement learning with augmented data*, NeurIPS 2020.
[1] Fujimoto, S. et al., *For sale: State-action representation learning for deep RL*, NeurIPS 2023.
[2] Chentanez, N. et al., *Intrinsically motivated reinforcement learning*, NeurIPS 2004.
[3] Quang, N. M. & Lauw, H. W., *Augmenting decision with hypothesis in reinforcement learning*, ICML 2024.
[4] Shen, J. & Yang, L. F., *Theoretically principled deep RL acceleration via nearest neighbor function approximation*, AAAI 2021.

---

### Meta-Review · Area_Chair_ahEF · 2026-01-01

**Summary:**

Some reviewers raised significant concerns regarding the paper’s novelty and the necessity of the proposed modules, which made this method complicated without sufficient justification. In addition, the theoretical analysis was not insightful, with several propositions being identified as either non-rigorous or standard results from existing literature. After reading the paper, comments and rebuttal, I find that the core concerns raised by the reviewers remain unresolved. Hence, I recommend rejection.

**Reviewer Concerns:**

Reviewer NHmE: This reviewer questioned the paper’s novelty and the necessity of the proposed modules, noting that the AVPD module may hinder exploration, the Q-value prior distillation loss appears unstable, and it lacks comparison to similar existing methods like LAPO and A2PO. While the authors provided a response, I find that they did not successfully address the requested comparison with LAPO and A2PO.

Reviewer H77H: This reviewer identified several sensitivities including $Q$ and $H$, and asked for more backbones. Most of these concerns are addressed.

Reviewer vq3Q: This reviewer raised concerns regarding methodological motivation, comparative positioning, and theoretical depth. After the first round of rebuttal, this reviewer maintained that the main cocerns were not sufficiently addressed. After the second round of rebuttal, I find that these core issues remain unresolved.

Reviewer eZE4: This reviewer challenged the fundamental necessity of the method, and state that the method is overcomplicated and not principled. In addition, several propositions seem to be irrelevant to the method and that multiple algorithmic components remain unclear. Despite an extensive discussion between the authors and the reviewer, these significant concerns have not been fully addressed.

**Reviewer Scores:**

Reviewer H77H increased the score from 4 to 6, while I think other reviewers would keep the original score.

---

### Decision · Program_Chairs · 2026-01-26

Reject